# Decomposition of one-layer neural networks via the infinite sum of reproducing kernel Banach spaces

## Abstract

In this paper, we define the sum of RKBSs using the characterization theorem of RKBSs and show that the sum of RKBSs is compatible with the direct sum of feature spaces. Moreover, we decompose the integral RKBS $\mathcal{F}_\sigma(\mathcal{X}, \Omega)$ into the sum of $p$-norm RKBSs $\{\mathcal{L}_\sigma^1(\mu_i)\}_{i \in I}$. Finally, we provide some applications to enhance the structural understanding of the integral RKBS class.

## 1 Introduction

To analyze the performance of neural networks, the hypothesis space represented by (infinite width) neural networks has been studied. Based on the concept of variation spaces (Kurková & Sanguineti, 2001; Mhaskar, 2004), Bach (2017) defined the $\mathcal{F}_1$ spaces as an integral representation of the neural networks using the total variation norm. In subsequent, E & Stephan (2022) defined the Barron spaces employing the path norm and showed that the $\mathcal{F}_1$ spaces and the Barron spaces can be isometrically isomorphic when using the Rectified Linear Unit (RELU) activation function.

The concept of Reproducing Kernel Banach Spaces (RKBSs) is a generalization of the Reproducing Kernel Hilbert Spaces (RKHSs), similar to how Banach spaces extend Hilbert spaces (Zhang et al., 2009). Relating to neural networks, Bartolucci et al. (2023) defined a class of integral RKBSs which are variants of the $\mathcal{F}_1$ spaces. They defined a class of integral RKBSs through the characterization theorem of the RKBS introduced by Combettes et al. (2018) which describe an RKBS using a feature space and its associated feature map.

In this study, our primary focus is on a class of integral RKBSs. We aim to decompose this function space and identify its fundamental building blocks. Decomposing a function space entails preserving both its algebraic operations and topological properties. Since we are dealing with RKBS, we additionally need to ensure that the decomposition preserves the property that evaluation functionals remain continuous (see Definition 3.2). Considering the case of RKHS, there exists a sum of RKHSs that naturally extends the space in a canonical manner, resulting in an RKHS (Aronszajn, 1950). Using this approach, we aim to define the (potentially infinite) sum of RKBSs and investigate its relationship with the feature spaces.

The main questions of this paper are the following:

(1) Finding a natural definition for the sum of RKBSs that is compatible with the usual direct sum of Banach spaces.

(2) How can we decompose a class of integral RKBSs using the sum defined in question (1)?

To answer these questions, we define the sum of RKBSs, see Proposition 3.7, and show that the direct sum of the feature (Banach) spaces is compatible with the sum of RKBSs (Proposition 4.2). Roughly speaking, it is well-known that the space of the Radon measures can be decompose as the vast $l^1$ direct sum of $L^1$ spaces. As an analogue of the fact described above, we decompose the class of integral RKBSs using the sum of p-norm RKBSs (Theorem 4.4).

## 1.1 RELATED WORK

Before the era of neural networks, one of the main topics in machine learning was the kernel method, exemplified by concepts such as Reproducing Kernel Hilbert Spaces (RKHSs) and Support Vector Machines (Aronszajn, 1950; Steinwart & Christmann, 2008; Berlinet & Thomas-Agnan, 2011). Machine learning models that use RKHS as their hypothesis space are guaranteed the existence of a solution through the Representer Theorem. Additionally, an algorithm for explicitly finding this solution is clearly presented (Smola & Schölkopf, 1998; Shalev Shwartz & Ben David, 2014). This characteristic significantly reduces the gap between theoretical understanding and practical application. One standard method for extending RKHS is through their sum, which plays a crucial role in enhancing the approximation ability of machine learning models. For instance, approaches like the multiple kernel algorithm demonstrate the utility of such extensions in effectively capturing diverse features of data (Yamanishi et al., 2004; Gönen & Alpaydın, 2011).

However, RKHS-based learning algorithms exhibit certain limitations due to their inner product structure. To address these challenges, the concept of Reproducing Kernel Banach Spaces (RKBSs) was introduced. Numerous studies have explored its theoretical foundations and applications (Zhang et al., 2009; Song et al., 2013; Fasshauer et al., 2015; Lin et al., 2022). Meanwhile, early theoretical research on neural networks primarily focused on approximation properties (Cybenko, 1989; Hornik et al., 1990; Barron, 1993). This line of inquiry led to further investigations into the hypothesis spaces of infinitely wide neural networks, culminating in the introduction of concepts such as Barron spaces and variation spaces(Bach, 2017; E et al., 2022; E & Stephan, 2022; Siegel & Xu, 2023).

Recent studies have attempted to analyze the hypothesis spaces of neural networks within the RKBS framework. For this purpose, the concept of integral RKBS has been introduced, which enables the proof of the Representer Theorem for one-layer neural networks (Bartolucci et al., 2023). However, unlike RKHS-based models, neural networks lack a clear algorithm for finding the solutions guaranteed by the Representer Theorem. In this study, we propose a method to decompose the hypothesis space of one-layer neural networks while preserving the RKBS structure. This approach enables a bottom-up exploration of the hypothesis space of one-layer neural networks, with the goal of contributing to the development of explicit algorithms for solutions guaranteed by the Representer Theorem in neural network settings.

## 1.2 ORGANIZATION

This paper is organized as follows. In Section 2, we briefly review the definitions and basic facts of the functional analysis. In Section 3, following Bartolucci et al. (2023); Spek et al. (2022), we introduce the definition of RKBSs and related function subclasses, namely a class of integral RKBSs and a class of p-norm RKBSs. We present some basic properties of these function classes, particularly focusing on the comparison between integral RKBSs and spaces of continuous functions (Proposition 3.5). Moreover, we define the sum of RKBSs, which is a modified version of Example 3.13 in Combettes et al. (2018) and the theorem in 353p of Aronszajn (1950), by using the characterization theorem of an RKBS. In Section 4, we state the main result of this article. We provide the compatibility between the sum of RKBSs and the direct sum of feature (Banach) spaces. Furthermore, using the compatibility (Proposition 4.2), we obtain that a class of integral RKBSs can be decomposed into the sum of p-norm RKBSs (Theorem 4.4). In Section 5, we provide direct applications of Theorem 4.4, showing how the size of the RKBS $\mathcal{F}_\sigma(\mathcal{X}, \Omega)$ compares to the finite sum of p-norm RKHSs.

## 2 PRELIMINARIES AND NOTATIONS

In this paper, we denote $I$ as a non-empty index set and the set $\{1, \ldots, n\}$ is denoted by $[n]$. We consistently use $p$ and $q$ as conjugate indices, where $p$ satisfies $1 \leq p < \infty$. The data space is represented as $\mathcal{X}$, and the parameter space as $\Omega$. For convenience, we assume that $\mathcal{X}$ and $\Omega$ are compact subsets of $\mathbb{R}^d$ and $\mathbb{R}^D$ for some $d, D \in \mathbb{N}$, respectively. Furthermore, we use the notation $\cong$ to denote isomorphisms between vector spaces and the notation $\cong_{\mathcal{B}}$ to denote isometric isomorphisms between Banach spaces.

## 2.1 Direct Sum of Normed Vector Spaces

Let $\{a_i\}_{i \in I}$ be a family of elements in a Hausdorff commutative topological group (HCTG) $H$. Define $\mathcal{F}$ as the collection of all finite subsets of $I$, and order $\mathcal{F}$ by inclusion. Then $\mathcal{F}$ becomes a directed set. For each $F \in \mathcal{F}$, define $a_F := \sum_{i \in F} a_i$. Since $F$ is a finite set, $a_F$ would be well-defined. Thus, $(a_F)_{F \in \mathcal{F}}$ is a net in $H$. The family $\{a_i\}_{i \in I}$ is said to be summable if the net $(a_F)_{F \in \mathcal{F}}$ converges. In this case, the limit is called the sum of the family $\{a_i\}_{i \in I}$, and we denote it by $_H\sum_{i \in I} a_i$. When we consider sums in the norm topology of $\mathbb{R}$, we use the term $\sum_{i \in I} a_i$ instead of $_{\mathbb{R}}\sum_{i \in I} a_i$. The contents related to the summable family in HCTG and $\mathbb{R}$ can be found in III §5 and IV §7 of Bourbaki (1971) respectively.

For a given index set $I \neq \emptyset$, let $\{X_i\}_{i \in I}$ be a family of sets indexed by $I$. Then the direct product of the sets in $\{X_i\}_{i \in I}$ is defined by $\prod_{i \in I} X_i := \{\mathbf{x} : I \to \bigcup_{i \in I} X_i : \mathbf{x}(i) \in X_i \text{ for all } i \in I\}$. When we assume that $X_i \neq \emptyset$ for all $i \in I$, by the axiom of choice, $\prod_{i \in I} X_i$ is the non-empty set. In this case, for $j \in I$, we can define $p_j : \prod_{i \in I} X_i \to X_j$ by $p_j(\mathbf{x}) = \mathbf{x}(j)$ for $\mathbf{x} \in \prod_{i \in I} X_i$. And we call $p_j$ is the $j$-th canonical projection. By abuse of notation, for any $\mathbf{x} \in \prod_{i \in I} X_i$, we denote $\mathbf{x}$ by $(x_i)_{i \in I}$ which means $\mathbf{x}(j) = x_j \in X_j$ for all $j \in I$. When $\{X_i\}_{i \in I}$ is a collection of $\mathbb{R}$-vector spaces, the direct product of $\{X_i\}_{i \in I}$ is the vector space $\prod_{i \in I} X_i$ with componentwise addition and scalar multiplication. In this case, the canonical projections are linear maps. Furthermore, if $\{X_i\}_{i \in I}$ are topological spaces, then we can define the direct product of $\{X_i\}_{i \in I}$ by giving a topology on $\prod_{i \in I} X_i$, called the product topology. Under this situation, the canonical projections are continuous maps.

Let $\{X_i : i \in I\}$ be a collection of normed vector spaces indexed by $I$. Then we can define the direct sum of the normed vector spaces $\{X_i : i \in I\}$ as follows:

**Definition 2.1** (The direct sum of normed vector spaces (Conway, 1997)). *For $1 \leq p < \infty$, we define*

$$\bigoplus_{i \in I}^{p} X_i := \left\{\mathbf{x} \in \prod_{i \in I} X_i : \left[\sum_{i \in I} \|\mathbf{x}(i)\|_{X_i}^{p}\right]^{\frac{1}{p}} < \infty\right\}$$

*as a normed vector space equipped with the norm $\|\mathbf{x}\|_{\bigoplus_{i \in I}^{p} X_i} = \left[\sum_{i \in I} \|\mathbf{x}(i)\|_{X_i}^{p}\right]^{\frac{1}{p}}$. For $p = \infty$, we define*

$$\bigoplus_{i \in I}^{\infty} X_i := \left\{\mathbf{x} \in \prod_{i \in I} X_i : \sup_{i \in I} \|\mathbf{x}(i)\|_{X_i} < \infty\right\}$$

*as a normed vector space equipped with the norm $\|\mathbf{x}\|_{\bigoplus_{i \in I}^{\infty} X_i} = \sup_{i \in I} \|\mathbf{x}(i)\|_{X_i}$.*

In particular, if each $X_i$ is a Banach space, then the direct sum of $\{X_i\}_{i \in I}$ is a Banach space. Let $p$ and $q$ be conjugate indices with $1 \leq p < \infty$. We can obtain the following relationship between the duality and the direct sum (see III §5 Exercise 4 in Conway (1997)):

$$\text{Define the map } \Phi : \bigoplus_{i \in I}^{q} (X_i^*) \to \left(\bigoplus_{i \in I}^{p} X_i\right)^* \text{ as } \Phi\left((g_i)_{i \in I}\right)(f_i)_{i \in I} = \sum_{i \in I} \langle g_i, f_i \rangle \qquad (2.1)$$

for $(g_i)_{i \in I} \in \bigoplus_{i \in I}^{q} (X_i^*)$ and $(f_i)_{i \in I} \in \bigoplus_{i \in I}^{p} X_i$. Then $\Phi$ is well-defined and it is an isometric isomorphism.

## 2.2 Review of Measure Theory

Let $K$ be a compact metric space. Then we know that the Borel and Baire $\sigma$-algebra over $K$ is coincide and every Borel measure on $K$ is Radon measure (see Proposition 6.3.4 and Theorem 7.1.7 in Bogachev & Ruas (2007)). Let $C(K)$ be the Banach space consisting of continuous real-valued functions defined on $K$, equipped with the supremum norm. We denote by $\mathcal{M}(K)$ the Banach space of (signed) Borel measures defined on $K$, endowed with the total variation norm. Additionally, the set of positive measures in $\mathcal{M}(K)$ is denoted by $\mathcal{M}(K)^+$, and the set of probability measures

in $\mathcal{M}(K)$ is denoted by $P(K)$. We know that by the Riesz Representation Theorem, there is an isometric isomorphism

$$\Lambda : \mathcal{M}(K) \to C(K)^* \text{ defined by } \Lambda(\mu)(f) := \int_K f d\mu \quad \text{for } \mu \in \mathcal{M}(K) \text{ and } f \in C(K). \quad (2.2)$$

Let a measure space $(K, \Sigma, \mu)$ be given. Then, for $1 \leq p \leq \infty$, we can define the Banach space $L^p(\mu)$ consisting of equivalence class of $p$-th power integrable functions with norm
$$\begin{cases} \|f\|_p = \left(\int_K |f| d\mu\right)^{1/p} < \infty, & \text{if } 1 \leq p < \infty \\ \|f\|_\infty = \operatorname{ess\,sup}|f| < \infty, & \text{if } p = \infty \end{cases}. \text{ When } p \text{ and } q \text{ are conjugate indices with } 1 < p < \infty, \text{ there is an isometric isomorphism}$$

$$\Xi : L^q(\mu) \to L^p(\mu)^* \text{ defined by } \Xi(g)(f) := \int_K f g d\mu \quad \text{for } g \in L^q(\mu) \text{ and } f \in L^p(\mu). \quad (2.3)$$

It is also true for the case of $p = 1, q = \infty$ if the measure space $(K, \Sigma, \mu)$ is indeed $\sigma$-finite. We use the notation $L^p(K, \mu)$ instead of $L^p(\mu)$ if there is a need to distinguish the domain space $K$.

A family $\mathfrak{F}$ of measures in $\mathcal{M}(K)^+$ is called singular if $\mu \perp \nu$ whenever $\mu, \nu \in \mathfrak{F}$ and $\mu \neq \nu$ (see Definition 4.2.4 and Definition 4.6.1 in Dales et al. (2016)). Let $\mathfrak{S}$ be a nonempty subset of $\mathcal{M}(K)^+$. Then, by Zorn's lemma, there exists a maximal element in the set $\{\mathfrak{A} : \mathfrak{A} \subset \mathfrak{S}, \mathfrak{A} \text{ is a singular family in } \mathcal{M}(K)^+\}$. This maximal element is called a maximal singular family in $\mathfrak{S}$. Let $\{\mu_i\}_{i \in I}$ be a maximal singular family in $P(K)$. Then, there exists an isometric isomorphism

$$\Theta : \bigoplus_{i \in I}^{1} L^1(\mu_i) \to \mathcal{M}(K) \text{ defined by } \Theta\left((f_i)_{i \in I}\right) =_{\mathcal{M}(K)} \sum_{i \in I} \rho_i \quad (2.4)$$

for $(f_i)_{i \in I} \in \bigoplus_{i \in I}^{1} L^1(\mu_i)$, where $\rho_i(B) = \int_B f_i d\mu_i$ for all $i \in I$ and Borel set $B$ in $K$. (see Theorem 4.6.6 in Dales et al. (2016) and Proposition 4.3.8 in Albiac & Kalton (2016)). We use the notation $\Phi, \Lambda, \Xi$ and $\Theta$ liberally in situations that are isometrically isomorphic, as described above.

# 3 REPRODUCING KERNEL BANACH SPACES

## 3.1 DEFINITION OF RKBS

When we consider $\mathbb{R}^{\mathcal{X}} = \prod_{x \in \mathcal{X}} \mathbb{R}_x$, where $\mathbb{R}_x$ is just a copy of $\mathbb{R}$ for each $x \in \mathcal{X}$, there is a natural topological structure called the product topology. Equivalently, it is the initial topology with respect to the family of canonical projections $\{p_x : \mathbb{R}^{\mathcal{X}} \to \mathbb{R}_x\}_{x \in \mathcal{X}}$. Since this topology is compatible with the vector space structure of $\mathbb{R}^{\mathcal{X}}$, $\mathbb{R}^{\mathcal{X}}$ becomes a Hausdorff topological vector space (HTVS). Thus, we may consider a summable family $(a_i)_{i \in I}$ in $\mathbb{R}^{\mathcal{X}}$ and denote its sum in $\mathbb{R}^{\mathcal{X}}$ by $_{\mathbb{R}^{\mathcal{X}}} \sum_{i \in I} a_i$ if it exists.

Let $V$ be a linear subspace of $\mathbb{R}^{\mathcal{X}}$. Then a topology on $V$ induced by the product topology of $\mathbb{R}^{\mathcal{X}}$ again gives $V$ the structure of a HTVS. Additionally, due to the transitivity of the initial topology, the subspace topology on $V$ coincides with the initial topology induced by the family of restrictions $\{p_x|_V : V \to \mathbb{R}_x\}_{x \in \mathcal{X}}$. We denote such a HTVS as $(V, \{p_x|_V\}_{x \in \mathcal{X}})$. (Relating reference can be found in Narici & Beckenstein (2010); Bogachev & Smolyanov (2017); Bourbaki (1971)). To distinguish between an index set $I$ and the data set $\mathcal{X}$, we use the term for the case of the latter as follows:

**Definition 3.1.** *Let $V$ be a linear subspace of $\mathbb{R}^{\mathcal{X}}$. For each $x \in \mathcal{X}$, we use the term **evaluation functional** at $x \in \mathcal{X}$ **on** $V$ to refer to the restriction of the canonical projection $p_x|_V : V \to \mathbb{R}_x$, denoting it as $ev_x$. Specifically, the function $ev_x : V \to \mathbb{R}$ is a linear functional defined by $ev_x(f) = f(x)$ for all $f \in V$.*

Now we define a reproducing kernel Banach space on $\mathcal{X}$ as follows:

**Definition 3.2** (Definition of reproducing kernel Banach space (Bartolucci et al., 2023; Lin et al., 2022))**.** *For a given set $\mathcal{X}$, a **reproducing kernel Banach space (RKBS)** $\mathcal{B}$ on $\mathcal{X}$ is a Banach space $\mathcal{B}$ of functions $f : \mathcal{X} \to \mathbb{R}$ such that*

1. *as a vector space, $\mathcal{B}$ is a linear subspace of $\mathbb{R}^{\mathcal{X}}$*

2. *for all $x \in \mathcal{X}$, there is a constant $C_x \geq 0$ such that for all $f \in \mathcal{B}$, $|f(x)| \leq C_x \|f\|_{\mathcal{B}}$.*

According to the definition, all evaluation functionals on $\mathcal{B}$ are continuous. In other words, we have that for all $x \in \mathcal{X}$, $ev_x \in \mathcal{B}^*$. Therefore, the norm topology of an RKBS $(\mathcal{B}, \|\cdot\|_{\mathcal{B}})$ is finer than the HTVS $(\mathcal{B}, \{ev_x\}_{x \in \mathcal{X}})$. Let $(\mathcal{B}, \|\cdot\|_1)$ and $(\mathcal{B}, \|\cdot\|_2)$ be two RKBSs on the same linear subspace $\mathcal{B}$ of $\mathbb{R}^{\mathcal{X}}$. Then by the Closed Graph Theorem, two norms $\|\cdot\|_1$ and $\|\cdot\|_2$ on the linear space $\mathcal{B}$ is equivalent (see I §3 Exercise 2 in Bourbaki (1953) and Corollary $\text{IV}_1$ in Aronszajn (1950))[1]. In other words, when we have a function space $\mathcal{B}$, we can give an unique RKBS structure on $\mathcal{B}$ up to equivalence of norms.

We will consider these RKBSs as hypothesis spaces in machine learning. The reason for using RKBS is as follows: When defining a hypothesis space (or function space) in machine learning, we consider completeness and pointwise convergence as the minimal assumptions required for the properties of the function space (see Chapter 1 in Berlinet & Thomas-Agnan (2011)).

### 3.2 CHARACTERIZATION OF RKBSS

Before we state the characterization theorem of RKBSs, we introduce a method that induces a mathematical structure from a pre-existing structure. Let $V$ be a normed vector space over $\mathbb{R}$ equipped with the norm $\|\cdot\|_V$, and let $W$ be a vector space over $\mathbb{R}$. If there is a vector space isomorphism $T : V \to W$, then $\|T^{-1}(\cdot)\|_V : W \to \mathbb{R}$ defines a norm on $W$. Furthermore, when we consider $W$ as a normed vector space equipped with the norm $\|T^{-1}(\cdot)\|_V$, the linear isomorphism $T : (V, \|\cdot\|_V) \to (W, \|T^{-1}(\cdot)\|_V)$ becomes an isometric isomorphism (This is referred to as the transport of structure).

Let $V$ and $W$ be vector spaces. If $T : V \to W$ is a linear map, then there exists an unique linear map $\hat{T} : V/\ker T \to W$ such that $\hat{T} \circ \pi = T$, where $\pi : V \to V/\ker T$ defined by $\pi(v) = [v]$ for $v \in V$. Throughout this paper, we use the notation $\hat{T}$ to denote the induced linear map described above in similar situations. We now state the characterization theorem of RKBSs introduced by Combettes et al. (2018).

**Theorem 3.3** (Characterization of RKBSs (Bartolucci et al., 2023; Combettes et al., 2018))**.** *A linear subspace $\mathcal{B}$ of $\mathbb{R}^{\mathcal{X}}$ is an RKBS on $\mathcal{X}$ if and only if there exists a Banach space $\Psi$ and a map $\psi : \mathcal{X} \to \Psi^*$ such that $\mathcal{B} = \text{im}(A) = \{f : \exists \nu \in \Psi \text{ s.t. } A(\nu) = f\}$ with the norm $\|f\|_{\mathcal{B}} = \inf_{\nu \in A^{-1}(f)} \|\nu\|_{\Psi}$,*

*where $A : \Psi \to \mathbb{R}^{\mathcal{X}}$ is a linear map defined by $(A\nu)(x) := \langle \psi(x), \nu \rangle$ for $x \in \mathcal{X}$ and $\nu \in \Psi$.*

Note that the linear map $A$ is the linear transformation induced from the family of the linear maps $\{\psi(x) : \Psi \to \mathbb{R}_x\}_{x \in X}$ by the universal property of the direct product of the vector spaces $\{\mathbb{R}_x\}_{x \in \mathcal{X}}$. We briefly review the proof provided in Bartolucci et al. (2023). In the necessity part of the proof, it is shown that $\ker A$ is closed in $\Psi$ by the following equations:

$$\ker(A) = \{\nu \in \Psi : \psi(x)(\nu) = 0 \text{ for all } x \in \mathcal{X}\} = \bigcap_{x \in X} \ker \psi(x). \tag{3.1}$$

Thus, $\Psi/\ker A$ can be a Banach space with the quotient norm. Consider the linear map $\hat{A} : \Psi/\ker A \to \mathbb{R}^{\mathcal{X}}$ such that $A = \hat{A} \circ \pi$. Since $\hat{A} : \Psi/\ker A \cong \text{im}(A)$ is an isomorphism of vector spaces, by the transport of the structure, $\mathcal{B} = \text{im}(A)$ becomes a Banach space with the norm:

$$\|f\|_{\mathcal{B}} = \|\hat{A}^{-1}(f)\|_{\Psi/\ker(A)} = \inf_{\nu \in \pi^{-1}(\hat{A}^{-1}(f))} \|\nu\|_{\Psi} = \inf_{\nu \in A^{-1}(f)} \|\nu\|_{\Psi}$$

The evaluation functionals are continuous as follows: for any $f \in \mathcal{B}$ and $\nu \in A^{-1}(f)$, we have $|f(x)| = |A\nu(x)| \leq \|\psi(x)\|_{\Psi^*} \|\nu\|_{\Psi}$. Thus, we can deduce that for all $x \in \mathcal{X}$,

$$\|ev_x(f)\|_{\mathbb{R}} = |f(x)| \leq \|\psi(x)\|_{\Psi^*} \inf_{\nu \in A^{-1}(f)} \|\nu\|_{\Psi} = \|\psi(x)\|_{\Psi^*} \|f\|_{\mathcal{B}}. \tag{3.2}$$

From now on, for a given RKBS $\mathcal{B}$, we consider a corresponding space $\Psi$, a map $\psi$ and an induced linear map $A$. In this situation, by abuse of notation, we may say that an RKBS triple $\mathcal{B} = (\Psi, \psi, A)$ is given. Each component of the triple $(\Psi, \psi, A)$ has a specific name. Specifically, we refer to $\Psi$ as a feature space, $\psi$ as a feature map, and $A$ as an RKBS map in order.

---

[1] https://terrytao.wordpress.com/tag/weak-topology/

### 3.3 ONE-LAYER NEURAL NETWORKS

In this subsection, we assume that $\Omega_1 \subset \mathbb{R}^d$ and $\Omega_2 \subset \mathbb{R}$ are compact, and let $\Omega = \Omega_1 \times \Omega_2$. Consider a continuous nonlinear function $g : \mathbb{R} \to \mathbb{R}$. The prediction function represented by a one-layer neural network with a one-dimensional target can be expressed as follows:

$$f(x) = \sum_{i=1}^{m} \eta_i g(x \cdot \theta_i - b_i), \tag{3.3}$$

where $x \in \mathcal{X}$, $\theta_i \in \Omega_1$, $b_i \in \Omega_2$ and $\eta_i \in \mathbb{R}$ for $i = 1, \ldots, m$. For convention, and with some abuse of notation, we define a continuous function $\sigma : \mathcal{X} \times \Omega \to \mathbb{R}$ by $\sigma(x, w) = g(x \cdot \theta - b)$ where $w = (\theta, b)$. This gives the following simplified representation: $f(x) = \sum_{i=1}^{m} \eta_i \sigma(x, w_i)$. Using measure-theoretic notation, we can have an integral representation of the equation 3.3: $f(x) = \int_\Omega \sigma(x, w) d\left(\sum_{i=1}^{m} \eta_i \delta_{w_i}\right)$ where $\delta_{w_i}$ is the Dirac measure at $w_i$. When considering the limit as $m \to \infty$ in equation 3.3, we obtain the following:

$$\int_\Omega \sigma(x, w) d\left(\sum_{i=1}^{m} \eta_i \delta_{w_i}\right) \to \int_\Omega \sigma(x, w) d\mu(w),$$

for some $\mu \in \mathcal{M}(\Omega)$. A more detailed explanation can be found in Chapter 9 of Bach (2024). In the following subsection, we will define the hypothesis space of one-layer neural networks in a more abstract way using this relaxed expression.

### 3.4 INTEGRAL RKBS AND P-NORM RKBS

Directly using the characterization theorem, we can define the hypothesis spaces that are considered to represent one-layer neural networks. Until section 4, we consider a fixed element $\sigma$ in $C(\mathcal{X} \times \Omega)$, where $\mathcal{X}$ is a compact subset of $\mathbb{R}^d$ and $\Omega$ is a compact subset of $\mathbb{R}^D$ for some $d, D \in \mathbb{N}$.

Let $V$ and $W$ be a real normed vector spaces. If we denote $V^{**}$ be the bidual space of $V$, then there is a linear isometric embedding $\iota : V \to V^{**}$, called the canonical embedding of $V$ in $V^{**}$, defined by $\iota(v)(v^*) = v^*(v)$ for $v \in V$ and $v^* \in V^*$. For a given bounded linear operator $T : V \to W$, the dual operator of $T$ is the linear operator $T^* : W^* \to V^*$ defined by $T^*(w^*) := w^* \circ T$ for $w^* \in W^*$.

**Definition 3.4** (A class of integral RKBSs, associated with the function $\sigma$ (Bartolucci et al., 2023; Spek et al., 2022))**.** *Let $\mathcal{M}(\Omega)$ be a feature space. Consider a feature map $\psi : \mathcal{X} \to \mathcal{M}(\Omega)^* (\underset{\mathcal{B}}{\cong} C(\Omega)^{**})$ defined by $\psi(x) = \Lambda^*(\iota(\sigma(x, \cdot)))$ for all $x \in \mathcal{X}$, where $\iota : C(\Omega) \to C(\Omega)^{**}$ is the canonical embedding of $C(\Omega)$ in $C(\Omega)^{**}$ and $\Lambda^*$ is the dual operator of $\Lambda : \mathcal{M}(\Omega) \to C(\Omega)^*$, which is defined in equation 2.2. Then there is a linear map $A : \mathcal{M}(\Omega) \to \mathbb{R}^\mathcal{X}$ defined by $(A\mu)(x) = \langle \psi(x), \mu \rangle = \int_\Omega \sigma(x, w) d\mu(w)$ for $x \in \mathcal{X}$ and $\mu \in \mathcal{M}(\Omega)$. An integral RKBS $\mathcal{F}_\sigma(\mathcal{X}, \Omega)$, associated with the function $\sigma$ is defined by the Banach space*

$$\mathcal{F}_\sigma(\mathcal{X}, \Omega) := \left\{ f \in \mathbb{R}^\mathcal{X} : \exists \mu \in \mathcal{M}(\Omega) \ s.t. \ \forall x \in \mathcal{X}, f(x) = \int_\Omega \sigma(x, w) d\mu(w) \right\}, \tag{3.4}$$

*equipped with the norm $\|f\|_{\mathcal{F}_\sigma(X, \Omega)} = \inf_{\mu \in A^{-1}(f)} \|\mu\|_{\mathcal{M}(\Omega)}$.*

In the above Definition 3.4, consider the linear map $A : \mathcal{M}(\Omega) \to \mathbb{R}^\mathcal{X}$. We deduce that, by the Dominated Convergence Theorem, $\text{im}(A)$ is a linear subspace of $C(\mathcal{X})$ (see Theorem 2.27 in Folland (1999)). Furthermore, from the inequality $\|A\mu\|_{C(\mathcal{X})} \leq \sup_{x \in \mathcal{X}, w \in \Omega} |\sigma(x, w)| \|\mu\|$, we can see that the map $A : \mathcal{M}(\Omega) \to C(\mathcal{X})$ is indeed a bounded operator. Recently, Steinwart showed that when $\mathcal{X}$ is an uncountable compact metric space, there is no RKHS $\mathcal{H}$ on $\mathcal{X}$ such that $C(\mathcal{X}) \subset \mathcal{H}$ (Steinwart, 2024). We can obtain a similar result for the class of integral RKBS as well.

**Proposition 3.5.** *The bounded operator $A : \mathcal{M}(\Omega) \to C(\mathcal{X})$ defined by*

$$(A\mu)(x) = \int_\Omega \sigma(x, w) d\mu(w)$$

*for $x \in \mathcal{X}$ and $\mu \in \mathcal{M}(\Omega)$ is compact.*

Using the proposition above, it follows that if $\operatorname{im}(A)$ is closed in $C(\mathcal{X})$, then $\operatorname{im}(A)$ has finite dimension. Thus, when $\mathcal{X}$ is an infinite compact metric space, we deduce that $\mathcal{F}_\sigma(\mathcal{X}, \Omega) \subsetneq C(\mathcal{X})$, and in general, $\mathcal{F}_\sigma(\mathcal{X}, \Omega)$ cannot be a Banach space if it equipped with the supremum norm.

**Definition 3.6** (A class of p-Norm RKBS, associated with the function $\sigma$ (Spek et al., 2022)). *Let $\pi \in P(\Omega)$ be given. Let $p$ and $q$ be conjugate indices such that $1 \leq p < \infty$. Take a feature space $\Psi$ as $L^p(\pi)$ and choose a feature map $\psi : \mathcal{X} \to (L^p(\pi))^*$ defined by $\psi(x) = \Xi(\sigma(x, \cdot))$ for $x \in \mathcal{X}$, where $\Xi : L^q(\pi) \to (L^p(\pi))^*$ is the isometric isomorphism defined in equation 2.3. Then, there is a linear map $A : L^p(\pi) \to \mathbb{R}^{\mathcal{X}}$ defined by $(Ah)(x) = \langle \psi(x), h \rangle$ for $x \in \mathcal{X}$ and $h \in L^p(\pi)$. We define a p-Norm RKBS $\mathcal{L}_\sigma^p(\pi)$, associated with the function $\sigma$ by the Banach space*

$$\mathcal{L}_\sigma^p(\pi) := \left\{ f \in \mathbb{R}^{\mathcal{X}} : \exists h \in L^p(\pi) \text{ s.t. } \forall x \in \mathcal{X}, f(x) = \int_\Omega \sigma(x, w) h(w) d\pi(w) \right\}, \tag{3.5}$$

*equipped with the norm $\|f\|_{\mathcal{L}_\sigma^p(\pi)} = \inf_{h \in A^{-1}(f)} \|h\|_{L^p(\pi)}$.*

When we consider the $p = 2$ case, we obtain the RKHS $\mathcal{L}_\sigma^2(\pi)$. This space corresponds to $\mathcal{F}_2$ as described in Bach (2017). The kernel of $\mathcal{L}_\sigma^2(\pi)$ is given by $k(x, y) = \int_\Omega \sigma(x, w) \sigma(y, w) d\pi(w)$ for $(x, y) \in \mathcal{X} \times \mathcal{X}$. Furthermore, $\mathcal{L}_\sigma^2(\pi)$ is embedded in $\mathcal{L}_\sigma^1(\pi)$ (that is, $\mathcal{L}_\sigma^2(\pi) \subset \mathcal{L}_\sigma^1(\pi)$) and for all $f \in \mathcal{L}_\sigma^2(\pi)$, $\|f\|_{\mathcal{L}_\sigma^1(\pi)} \leq \|f\|_{\mathcal{L}_\sigma^2(\pi)}$. As an analogue to the case of $L^p$ space, we sometimes use the notation $\mathcal{L}_\sigma^p(\Omega, \pi)$ instead of $\mathcal{L}_\sigma^p(\pi)$ to avoid confusion.

### 3.5 Infinite sum of reproducing kernel Banach spaces

Let an RKBS $\mathcal{B}$ be given. If we consider an evaluation functional on $\mathcal{B}$ evaluating at $x \in \mathcal{X}$ by $ev_x : \mathcal{B} \to \mathbb{R}$, as discussed earlier, then we have that for all $x \in \mathcal{X}$, $ev_x \in \mathcal{B}^*$. Thus, if we assume that a collection of RKBSs $\{\mathcal{B}_i\}_{i \in I}$ is given and denote $ev_x^i$ as the evaluation functional on $\mathcal{B}_i$ evaluating at $x \in \mathcal{X}$, then for all $i \in I$ and $x \in \mathcal{X}$, $ev_x^i \in \mathcal{B}_i^*$. Now, we define the sum of RKBSs as follows, modifying Example 3.13 in Combettes et al. (2018) and the theorem on page 353 of Aronszajn (1950):

**Proposition 3.7** (Infinite sum of reproducing kernel Banach spaces). *Let $p$ and $q$ be conjugate indices with $1 \leq p < \infty$. Let $\{\mathcal{B}_i\}_{i \in I}$ be a collection of RKBSs on $\mathcal{X}$. Suppose that for all $x \in \mathcal{X}$, $(ev_x^i)_{i \in I} \in \bigoplus_{i \in I}^q \mathcal{B}_i^*$. Let $\bigoplus_{i \in I}^p \mathcal{B}_i$ be a feature space and define a feature map $\mathbf{s} : \mathcal{X} \to \left(\bigoplus_{i \in I}^p \mathcal{B}_i\right)^*$ by $\mathbf{s}(x) = \Phi((ev_x^i)_{i \in I})$ for $x \in \mathcal{X}$, where $\Phi : \bigoplus_{i \in I}^q \mathcal{B}_i^* \to \left(\bigoplus_{i \in I}^p \mathcal{B}_i\right)^*$ is the isometric isomorphism defined in equation 2.1. Then there is a linear map $\mathcal{S} : \bigoplus_{i \in I}^p \mathcal{B}_i \to \mathbb{R}^{\mathcal{X}}$ defined by $\left(\mathcal{S}(f_i)_{i \in I}\right)(x) = \langle \mathbf{s}(x), (f_i)_{i \in I} \rangle$ for $(f_i)_{i \in I} \in \bigoplus_{i \in I}^p \mathcal{B}_i$ and $x \in \mathcal{X}$. By the Theorem 3.3, we can define an RKBS $\mathcal{B} = \operatorname{Im}(\mathcal{S}) = \{_{\mathbb{R}^{\mathcal{X}}} \sum_{i \in I} f_i : (f_i)_{i \in I} \in \bigoplus_{i \in I}^p \mathcal{B}_i\}$ equipped with the norm $\|f\|_{\mathcal{B}} = \inf_{(f_i)_{i \in I} \in \mathcal{S}^{-1}(f)} \|(f_i)_{i \in I}\|_{\bigoplus_{i \in I}^p \mathcal{B}_i} = \inf_{f = _{\mathbb{R}^{\mathcal{X}}} \sum_{i \in I} f_i} \|(f_i)_{i \in I}\|_{\bigoplus_{i \in I}^p \mathcal{B}_i}$.*

Note that the property of net in the initial topology implies that $f =_{\mathbb{R}^{\mathcal{X}}} \sum_{i \in I} f_i$ in $(\mathbb{R}^{\mathcal{X}}, \{p_x\}_{x \in \mathcal{X}})$ is equivalent to $f(x) = \sum_{i \in I} f_i(x)$ for all $x \in \mathcal{X}$. Thus, we have that

$$\mathcal{B} = \left\{ f \in \mathbb{R}^{\mathcal{X}} : \exists (f_i)_{i \in I} \in \bigoplus_{i \in I}^p \mathcal{B}_i \text{ s.t. } \forall x \in \mathcal{X}, f(x) = \sum_{i \in I} \langle ev_x^i, f_i \rangle \right\}$$

$$= \left\{ _{\mathbb{R}^{\mathcal{X}}} \sum_{i \in I} f_i : (f_i)_{i \in I} \in \bigoplus_{i \in I}^p \mathcal{B}_i \right\}.$$

From now on, we denote $\mathcal{B}$ mentioned in the Proposition 3.7 by $\sum_{i \in I}^p \mathcal{B}_i$ and call it the sum of the family of RKBSs $\{\mathcal{B}_i\}_{i \in I}$. In particular, for the case of $p = 1$, we denote $\mathcal{B}$ as $\sum_{i \in I} \mathcal{B}_i$. In the Proposition 3.7, we intentionally use the notation $\mathbf{s}$ for the feature map and $\mathcal{S}$ for the RKBS map to emphasize that they are used to represent the sum of RKBSs. Moreover, we denote the RKBS triple of the sum of RKBSs by $\sum_{i \in I}^p \mathcal{B}_i = (\bigoplus_{i \in I}^p \mathcal{B}_i, \mathbf{s}, \mathcal{S})$ using $\mathbf{s}$ and $\mathcal{S}$.

**Remark 3.8.** *Let a family of RKBS triples $\{\mathcal{B}_i = (\Psi_i, \psi_i, A_i)\}_{i \in I}$ be given. From the equation 3.2, we know that $\|ev_x^i\|_{B_i^*} \leq \|\psi_i(x)\|_{\Psi_i^*}$ for all $x \in \mathcal{X}$ and $i \in I$. Thus, instead of assuming $(ev_x^i)_{i \in I} \in \bigoplus_{i \in I}^q \mathcal{B}_i^*$ for all $x \in \mathcal{X}$, it suffices to assume that $(\psi_i(x))_{i \in I} \in \bigoplus_{i \in I}^q \Psi_i^*$ for all $x \in \mathcal{X}$.*

## 4 MAIN RESULTS

### 4.1 COMPATIBILITY BETWEEN THE SUM OF RKBSS AND THE DIRECT SUM OF FEATURE SPACES

In this section, we present the compatibility between the sum of RKBSs and the direct sum of feature spaces. Before stating our main proposition, we prove the following lemma, which says that the restriction to the direct sum of Banach spaces of the product of (quotient, isometry) maps preserves their properties.

**Lemma 4.1.** *Suppose $1 \le p < \infty$, and let a family of Banach spaces $\{X_i\}_{i \in I}$ be given.*

1. *Suppose that for each $i \in I$, $D_i$ is a closed linear subspace of $X_i$, and let $\pi_i : X_i \to X_i/D_i$ be the projection map defined by $\pi_i(x_i) := [x_i]$ for $x_i \in X_i$. Then, the map $\widetilde{(\pi_i)_{i \in I}} : \bigoplus_{i \in I}^p X_i \to \bigoplus_{i \in I}^p X_i/D_i$ defined by $\widetilde{(\pi_i)_{i \in I}}((x_i)_{i \in I}) = (\pi_i(x_i))_{i \in I}$ for $(x_i)_{i \in I} \in \bigoplus_{i \in I}^p X_i$ is a surjective bounded linear operator.*

2. *Assume there is another family of Banach spaces $\{Y_i\}_{i \in I}$. If for each $i \in I$, there is an isometric isomorphism $\phi_i : X_i \to Y_i$, then the map $\widetilde{(\phi_i)_{i \in I}} : \bigoplus_{i \in I}^p X_i \to \bigoplus_{i \in I}^p Y_i$ defined by $\widetilde{(\phi_i)_{i \in I}}((x_i)_{i \in I}) = (\phi_i(x_i))_{i \in I}$ for $(x_i)_{i \in I} \in \bigoplus_{i \in I}^p X_i$ is an isometric isomorphism.*

The following proposition is one of the main result of this paper. It states that when we have a family of RKBSs, there is an RKBS induced by the direct sum of feature spaces, which is isometrically isomorphic to the sum of the given family of RKBSs. Conversely, when we have an RKBS induced by the direct sum of feature spaces, there is a collection of RKBSs such that their sum is isometrically isomorphic to the given RKBS.

**Proposition 4.2** (Compatibility). *Let $I \ne \emptyset$ be an index set. Let $p$ and $q$ be conjugate indices, where $p$ satisfies $1 \le p < \infty$.*

1. *Suppose a family of RKBS triples $\{\mathcal{B}_i = (\Psi_i, \psi_i, A_i)\}_{i \in I}$ is given and $(ev_x^i)_{i \in I} \in \bigoplus_{i \in I}^q \mathcal{B}_i^*$ for all $x \in \mathcal{X}$. Then, there is an RKBS triple $\mathcal{B} = (\bigoplus_{i \in I}^p \Psi_i, \psi, A)$ such that $\mathcal{B} \underset{\mathcal{B}}{\cong} \sum_{i \in I}^p \mathcal{B}_i$.*

2. *For an RKBS triple $\mathcal{B} = (\bigoplus_{i \in I}^p \Psi_i, \psi, A)$, there is a family of reproducing kernel Banach spaces $\{\mathcal{B}_{i \in I}\}_{i \in I}$ such that $\mathcal{B} \underset{\mathcal{B}}{\cong} \sum_{i \in I}^p \mathcal{B}_i$.*

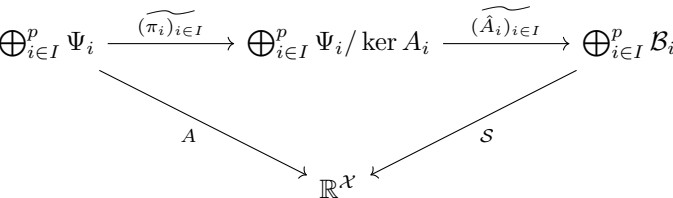

Figure 1: Commutative diagram for the compatibility

The diagram above intuitively illustrates the result we aim to demonstrate in Proposition 4.2. Detailed information about each map can be found in Appendix A.4.

### 4.2 DECOMPOSITION OF ONE-LAYER NEURAL NETWORKS

The following lemma shows that if there is an isometrically isomorphic feature space, then we can construct an isometrically isomorphic RKBS.

**Lemma 4.3.** *Let $\Psi_1$ be a Banach space and let $\mathcal{B}_2 = (\Psi_2, \psi_2, A_2)$ be an RKBS triple. If there is an isomeric isomorphism $\xi : \Psi_1 \to \Psi_2$, then there is an RKBS triple $\mathcal{B}_1 = (\Psi_1, \psi_1, A_1)$ such that $\mathcal{B}_1 \underset{\mathcal{B}}{\cong} \mathcal{B}_2$.*

Now, we introduce our main theorem. It state that the integral RKBS $\mathcal{F}_\sigma(\mathcal{X}, \Omega)$ defined in the Definition 3.4 can be decomposed into the sum of a family of p-norm RKBSs $\{\mathcal{L}_\sigma^1(\mu_i)\}_{i \in I}$ defined in the Definition 3.6, where $\{\mu_i\}_{i \in I}$ is a maximal singular family in $P(\Omega)$.

**Theorem 4.4.** *Let $\{\mu_i\}_{i \in I}$ be a maximal singular family in $P(\Omega)$. Then, we have the following:*

$$\mathcal{F}_\sigma(\mathcal{X}, \Omega) \underset{\mathcal{B}}{\cong} \sum_{i \in I} \mathcal{L}_\sigma^1(\mu_i).$$

**Remark 4.5.** *In the proof of the Theorem 4.4, we can see that the following set equality holds:*

$$\mathcal{F}_\sigma(\mathcal{X}, \Omega) = \sum_{i \in I} \mathcal{L}_\sigma^1(\mu_i).$$

*Furthermore, since $\Omega$ is a compact metric space in our setting, $\mathcal{L}_\sigma^1(\mu_i)$ is a separable RKBS for all $i \in I$. Thus, we decompose the integral RKBS $\mathcal{F}_\sigma(\mathcal{X}, \Omega)$ into infinitely many separable RKBSs.*

## 5 APPLICATION

Let $\{\mu_i\}_{i \in [n]}$ be any finite family in $P(\Omega)$. Since for each $i \in [n]$, $k_i(x, y) = \int_\Omega \sigma(x, w)\sigma(y, w)d\mu_i$ for $(x, y) \in \mathcal{X} \times \mathcal{X}$ is the reproducing kernel of $\mathcal{L}_\sigma^2(\mu_i)$, the sum kernel $k(x, y) = \sum_{i=1}^n k_i(x, y)$ is the reproducing kernel of $\sum_{i \in [n]}^2 \mathcal{L}_\sigma^2(\mu_i)$ (The notation $\sum_{i \in [n]}^2$ refers to the case where we defined it in Proposition 3.7 with $p = 2$ and $I = [n]$). In this setting, we can guarantee that $\sum_{i \in [n]}^2 \mathcal{L}_\sigma^2(\mu_i)$ is an RKHS. The following proposition shows that when we consider the finite singular family in $P(\Omega)$, the RKHS $\sum_{i \in [n]}^2 \mathcal{L}_\sigma^2(\mu_i)$ contained in the RKBS $\mathcal{F}_\sigma(\mathcal{X}, \Omega)$ with the same associated function $\sigma$.

**Proposition 5.1.** *For any finite singular family $\{\mu_i\}_{i \in [n]}$ in $P(\Omega)$, we have*

$$\sum_{i \in [n]}^2 \mathcal{L}_\sigma^2(\mu_i) \subset \mathcal{F}_\sigma(\mathcal{X}, \Omega).$$

Let a family of continuous functions $\{\sigma_i : \mathcal{X} \times \Omega \to \mathbb{R}\}_{i=1}^n$ be given. By the Tietze extension theorem and the pasting lemma, there is a continuous function $\sigma : \mathcal{X} \times \Omega \times [0, 1] \to \mathbb{R}$ which is a extension of the continuous function $\tilde{\sigma} : \mathcal{X} \times \Omega \times \{\frac{1}{n}, \ldots, \frac{n-1}{n}, 1\} \to \mathbb{R}$ defined by $\tilde{\sigma}(x, w, \frac{i}{n}) = \sigma_i(x, w)$ for all $i = 1, \ldots, n$, $x \in \mathcal{X}$ and $w \in \Omega$. In the following proposition, we show that the finite sum of p-norm RKBSs associated with different functions is contained in the integral RKBS associated with a suitable function when considering a larger parameter space. This means that the class of integral RKBSs is quite large due to its flexibility in choosing the dimension of the parameter space.

**Proposition 5.2.** *Let a family of continuous functions $\{\sigma_i : \mathcal{X} \times \Omega \to \mathbb{R}\}_{i=1}^n$ be given. Let $\{\pi_i\}_{i=1}^n$ be a collection of probability measures in $\Omega$. Then, we have*

$$\sum_{i \in [n]}^2 \mathcal{L}_{\sigma_i}^2(\Omega, \pi_i) \subset \mathcal{F}_\sigma(\mathcal{X}, \Omega \times [0, 1]).$$

**Remark 5.3.** *For the purpose of a realistic application, we consider the case where $p = 2$ in this section, but the results can also be generalized to the case where $1 \leq p < \infty$. Note that from the Corollary 13 in Spek et al. (2022), it is known that $\mathcal{F}_\sigma(\mathcal{X}, \Omega) = \bigcup_{\pi \in P(\Omega)} \mathcal{L}_\sigma^p(\pi)$. Thus, the Proposition 5.1 can be obtained without needing to consider the infinite sum of RKBSs. However, using this approach allows for a more systematic exploration.*

## 6 CONCLUSION AND FUTURE WORK

We showed that there is a compatibility property between the direct sum of feature spaces and the sum of RKBSs. By using this, we can decompose a class of integral RKBS $\mathcal{F}_\sigma(\mathcal{X}, \Omega)$ into the sum of p-norm RKBSs $\{\mathcal{L}_\sigma^1(\mu_i)\}_{i \in I}$. The advantage of this analytical method is that it allows for a

more structural understanding of the RKBS class through an appropriate decomposition approach. In Section 5, we partially explained these advantages by comparing the integral RKBS class to the previously known sum of RKHSs. Additionally, through these insights, we expect that it would be helpful in designing multiple kernel learning algorithms for the RKBS class. To ensure the feasibility of learning, we need to consider the Representer Theorem, which is discussed in paper Bartolucci et al. (2023) for the integral RKBS class. If the most generalized form of the Representer Theorem presented by Unser & Aziznejad (2022) can be extended to the infinite case, it seems likely that this would enable the recovery of the results obtained in Bartolucci et al. (2023) within the context of our findings.

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

## A  APPENDIX

### A.1  PROOF OF PROPOSITION 3.5

*Proof.* Since $\Omega$ is a compact metric space, $C(\Omega)$ is separable space. Let $\{\mu_n\}$ be a bounded sequence in $\mathcal{M}(\Omega) \cong_{\mathcal{B}} C(\Omega)^*$. Then, by the separable version of the Banach-Alaoglu Theorem, there exists a weak* convergent subsequence $\{\mu_{n_k}\}$ such that $\mu_{n_k} \xrightarrow{w^*} \mu$ (see Problem 10 of Chapter 4.9 in Kreyszig (1991)). Define $\Gamma := \{\sigma(x, \cdot) \in C(\Omega) : x \in \mathcal{X}\}$. Since $\Gamma$ is uniformly bounded

and pointwise equicontinuous, we have the following (see Exercise 8.10.134 in Bogachev & Ruas (2007)):

$$\lim_{n \to \infty} \|A\mu_{n_k} - A\mu\|_{C(\mathcal{X})} = \lim_{n \to \infty} \sup_{x \in \mathcal{X}} \left| \int \sigma(x, w) d(\mu_{n_k} - \mu)(w) \right|$$

$$= \lim_{n \to \infty} \sup_{f \in \Gamma} \left| \int f d(\mu_{n_k} - \mu) \right| = 0.$$

$\square$

### A.2 PROOF OF PROPOSITION 3.7

*Proof.* Let $\bigoplus_{i \in I}^p \mathcal{B}_i$ be a feature space and define a feature map $\mathbf{s} : \mathcal{X} \to \left(\bigoplus_{i \in I}^p \mathcal{B}_i\right)^*$ as $\mathbf{s}(x) = \Phi((ev_x^i)_{i \in I})$ for $x \in \mathcal{X}$, where $\Phi : \bigoplus_{i \in I}^q \mathcal{B}_i^* \to \left(\bigoplus_{i \in I}^p \mathcal{B}_i\right)^*$ is the isometric isomorphism defined in equation 2.1. Now, there is a linear transformation $\mathcal{S} : \bigoplus_{i \in I}^p \mathcal{B}_i \to \mathbb{R}^{\mathcal{X}}$ by $(\mathcal{S}(f_i)_{i \in I})(x) = \langle \mathbf{s}(x), (f_i)_{i \in I} \rangle$ for $(f_i)_{i \in I} \in \bigoplus_{i \in I}^p \mathcal{B}_i$ and $x \in \mathcal{X}$. Then, by the Theorem 3.3, $\bigoplus_{i \in I}^p \mathcal{B}_i / \ker(\mathcal{S}) = \text{im}(\mathcal{S})$ is an RKBS on $\mathcal{X}$ with the norm $\|f\|_{\mathcal{B}} = \inf_{(f_i)_{i \in I} \in \mathcal{S}^{-1}(f)} \|(f_i)_{i \in I}\|_{\bigoplus_{i \in I}^p \mathcal{B}_i}$. $\square$

### A.3 PROOF OF LEMMA 4.1

*Proof.* We know that for each $i \in I$, $\pi_i$ is a surjective bounded linear operator, and its norm satisfies $\|\pi_i\| \leq 1$ (see III §4 Theorem 4.2 in Conway (1997)). Additionally, there is an unique linear map $(\pi_i)_{i \in I} : \prod_{i \in I} X_i \to \prod_{i \in I} X_i / D_i$ such that $\pi_j \circ p_j = q_j \circ (\pi_i)_{i \in I}$ for all $j \in I$, where $p_j$ and $q_j$ are $j$-th canonical projections of $\prod_{i \in I} X_i$ and $\prod_{i \in I} X_i / D_i$, respectively. Consider the restriction of $(\pi_i)_{i \in I}$ to $\bigoplus_{i \in I}^p X_i$ and denote it by $\widetilde{(\pi_i)_{i \in I}} : \bigoplus_{i \in I}^p X_i \to \prod_{i \in I} X_i / D_i$. Let $(x_i)_{i \in I} \in \bigoplus_{i \in I}^p X_i$. Since $\widetilde{(\pi_i)_{i \in I}}((x_i)_{i \in I}) = (\pi_i(x_i))_{i \in I} \in \prod_{i \in I} X_i / D_i$ and $\sum_{i \in I} \|\pi_i(x_i)\|_{X_i / D_i}^p \leq \sum_{i \in I} \|x_i\|_{X_i}^p < \infty$, it follows that $\text{im}\left(\widetilde{(\pi_i)_{i \in I}}\right) \subset \bigoplus_{i \in I}^p X_i / D_i$. From this, we also know that $\widetilde{(\pi_i)_{i \in I}}$ is a bounded operator with norm less than 1.

It remains to show the surjectivity of $\widetilde{(\pi_i)_{i \in I}} : \bigoplus_{i \in I}^p X_i \to \bigoplus_{i \in I}^p X_i / D_i$. Let $(\pi_i(x_i))_{i \in I} \in \bigoplus_{i \in I}^p X_i / D_i$. Then, we have $\sum_{i \in I} \inf_{d_i \in D_i} \|x_i + d_i\|_{X_i}^p = \sum_{i \in I} (\inf_{d_i \in D_i} \|x_i + d_i\|_{X_i})^p = \sum_{i \in I} \|\pi_i(x_i)\|_{X_i / D_i}^p < \infty$ and the set $N = \{i \in I : \|\pi_i(x_i)\|_{X_i / D_i} > 0\}$ is countable. Let $f : \mathbb{N} \to N$ be a reordering bijection. From the definition of the infimum, for each $k \in \mathbb{N}$, we can take $\tilde{d}_{f(k)} \in D_{f(k)}$ such that

$$\|x_{f(k)} + \tilde{d}_{f(k)}\|_{X_{f(k)}}^p < \inf_{d_{f(k)} \in D_{f(k)}} \|x_{f(k)} + d_{f(k)}\|_{X_{f(k)}}^p + \frac{1}{k^2}.$$

Then, we have that:

$$\sum_{i \in N} \|x_i + \tilde{d}_i\|_{X_i}^p = \sum_{k=1}^{\infty} \|x_{f(k)} + \tilde{d}_{f(k)}\|_{X_{f(k)}}^p$$

$$< \sum_{k=1}^{\infty} \inf_{d_{f(k)} \in D_{f(k)}} \|x_{f(k)} + d_{f(k)}\|_{X_{f(k)}}^p + \sum_{k=1}^{\infty} \frac{1}{k^2} < \infty.$$

Thus, if we take $x_i' = \begin{cases} x_i + \tilde{d}_i & \text{if } i \in N, \\ 0 & \text{if } i \in I \setminus N \end{cases}$, then $(x_i')_{i \in I} \in \bigoplus_{i \in I}^p X_i$ and $\widetilde{(\pi_i)_{i \in I}}((x_i')_{i \in I}) = (\pi_i(x_i))_{i \in I}$. We can also prove the (2) directly.

$\square$

### A.4 PROOF OF PROPOSITION 4.2

*Proof.* From the Lemma 4.1, we know that there is a surjective bounded linear operator $\widetilde{(\pi_i)_{i \in I}} : \bigoplus_{i \in I}^p \Psi_i \to \bigoplus_{i \in I}^p \Psi_i / \ker A_i$ and an isometric isomorphism $\widetilde{(\hat{A}_i)_{i \in I}} : \bigoplus_{i \in I}^p \Psi_i / \ker A_i \to$

$\bigoplus_{i\in I}^{p} \mathcal{B}_i$. Let $\Phi : \bigoplus_{i\in I}^{q} \mathcal{B}_i^* \to \left(\bigoplus_{i\in I}^{p} \mathcal{B}_i\right)^*$ be the isometric isomorphism defined in equation 2.1. Since $(ev_x^i)_{i\in I} \in \bigoplus_{i\in I}^{q} \mathcal{B}_i^*$ for all $x \in \mathcal{X}$, we can apply the Proposition 3.7 to deduce that there is an RKBS triple for the summation of RKBSs $\sum_{i\in I}^{p} \mathcal{B}_i = (\bigoplus_{i\in I}^{p} \mathcal{B}_i, \mathbf{s}, \mathcal{S})$. Consider the map $A := \mathcal{S} \circ \widetilde{(\hat{A}_i)_{i\in I}} \circ \widetilde{(\pi_i)_{i\in I}} = \mathcal{S} \circ \widetilde{(A_i)_{i\in I}} : \bigoplus_{i\in I}^{p} \Psi_i \to \mathbb{R}^{\mathcal{X}}$. To verify the map $A$ is indeed an RKBS map, we show the following holds

$$(A(\mu_i)_{i\in I})(x) = \left(\mathcal{S}\left(\widetilde{(A_i)_{i\in I}}(\mu_i)_{i\in I}\right)\right)(x) = \left\langle \Phi((ev_x^i)_{i\in I}) \circ \widetilde{(A_i)_{i\in I}}, (\mu_i)_{i\in I} \right\rangle$$

for all $x \in \mathcal{X}$ and $(\mu_i)_{i\in I} \in \bigoplus_{i\in I}^{p} \Psi_i$. Thus, if we define a feature map $\psi : \mathcal{X} \to \left(\bigoplus_{i\in I}^{p} \Psi_i\right)^*$ by $\psi(x) = \Phi((ev_x^i)_{i\in I}) \circ \widetilde{(A_i)_{i\in I}} \in \left(\bigoplus_{i\in I}^{p} \Psi_i\right)^*$ for $x \in \mathcal{X}$, then we get an RKBS triple $\mathcal{B} = (\bigoplus_{i\in I}^{p} \Psi_i, \psi, A)$. Since $(\hat{A}_i)_{i\in I} \circ \widetilde{(\pi_i)_{i\in I}}$ is surjective, $\mathrm{im}(A) = \mathrm{im}(\mathcal{S})$ in terms of set equality. Also we note that, by the Theorem 3.3, $\mathcal{B} = \mathrm{im}(A)$ and $\sum_{i\in I}^{p} \mathcal{B}_i = \mathrm{im}(\mathcal{S})$ as sets. Since $\mathrm{im}(A)$ and $\mathrm{im}(\mathcal{S})$ both inherit the same algebraic structure from $\mathbb{R}^{\mathcal{X}}$, we can deduce that they are the same as vector space. The only remaining part of the proof is to show that for any $f \in \mathcal{B}$, $\|f\|_{\mathcal{B}} = \|f\|_{\sum_{i\in I}^{p} \mathcal{B}_i}$.

To prove (2), suppose that we have the RKBS triple $\mathcal{B} = (\bigoplus_{i\in I}^{p} \Psi_i, \psi, A)$. We denote $\Phi_0 : \bigoplus_{i\in I}^{q} \Psi_i^* \to \left(\bigoplus_{i\in I}^{p} \Psi_i\right)^*$ as the isometric isomorphism defined in equation 2.1. Since $\psi(x) \in \left(\bigoplus_{i\in I}^{p} \Psi_i\right)^*$ for all $x \in \mathcal{X}$, we know that

$$\Phi_0^{-1}(\psi(x)) \in \bigoplus_{i\in I}^{q} \Psi_i^*, \quad \|\Phi_0^{-1}(\psi(x))\|_{\bigoplus_{i\in I}^{q} \Psi_i^*} < \infty. \tag{A.1}$$

Now, we define for each $i \in I$, $\psi_i : \mathcal{X} \to \Psi_i^*$ by $\psi_i(x) = p_i(\Phi_0^{-1}(\psi(x)))$ for $x \in \mathcal{X}$, where $p_i$ is $i$-th canonical projection on $\prod_{i\in I} \Psi_i^*$. Then, for each $i \in I$, there is an RKBS map $A_i : \Psi_i \to \mathbb{R}^{\mathcal{X}}$ defined by $(A_i\mu_i)(x) = \langle \psi_i(x), \mu_i \rangle$ for $x \in \mathcal{X}$ and $\mu_i \in \Psi_i$. From the Theorem 3.3, we can get a family of RKBS triples $\{\mathcal{B}_i = (\Psi_i, \psi_i, A_i)\}_{i\in I}$. Let $\Phi : \bigoplus_{i\in I}^{q} \mathcal{B}_i^* \to \left(\bigoplus_{i\in I}^{p} \mathcal{B}_i\right)^*$ be the isometric isomorphism defined in equation 2.1. By the above equation A.1, we can deduce that $(\psi_i(x))_{i\in I} \in \bigoplus_{i\in I}^{q} \Psi_i^*$. Thus, the Remark 3.8 implies the existence of an RKBS triple for the sum of RKBSs $\sum_{i\in I}^{p} \mathcal{B}_i = (\bigoplus_{i\in I}^{p} \mathcal{B}_i, \mathbf{s}, \mathcal{S})$. From the following series of equations, we can see that $A = \mathcal{S} \circ \widetilde{(\hat{A}_i)_{i\in I}} \circ \widetilde{(\pi_i)_{i\in I}}$. For $x \in \mathcal{X}$ and $(\mu_i)_{i\in I} \in \bigoplus_{i\in I}^{p} \Psi_i$, we have that

$$(A((\mu_i)_{i\in I}))(x) = \langle \psi(x), (\mu_i)_{i\in I} \rangle = \langle \Phi_0(\Phi_0^{-1}(\psi(x))), (\mu_i)_{i\in I} \rangle = \sum_{i\in I} \langle p_i(\Phi_0^{-1}(\psi(x))), \mu_i \rangle$$

$$= \sum_{i\in I} \langle \psi_i(x), \mu_i \rangle = \langle \Phi((ev_x^i)_{i\in I}), (A_i\mu_i)_{i\in I} \rangle = (\mathcal{S}((A_i\mu_i)_{i\in I}))(x)$$

$$= \left(\mathcal{S}\left(\widetilde{(A_i)_{i\in I}}((\mu_i)_{i\in I})\right)\right)(x) = \left(\left(\mathcal{S} \circ \widetilde{(\hat{A}_i)_{i\in I}} \circ \widetilde{(\pi_i)_{i\in I}}\right)((\mu_i)_{i\in I})\right)(x).$$

For similar reasons as the previous case, we only need to prove that for any $f \in \mathcal{B}$, $\|f\|_{\mathcal{B}} = \|f\|_{\sum_{i\in I}^{p} \mathcal{B}_i}$.

We start by exploring the definition of each norm. The norm on the RKBS $\sum_{i\in I}^{p} \mathcal{B}_i$ is given by

$$\|f\|_{\sum_{i\in I}^{p} \mathcal{B}_i}^{p} = \inf\left\{\|(f_i)_{i\in I}\|_{\bigoplus_{i\in I}^{p} \mathcal{B}_i}^{p} : (f_i)_{i\in I} \in \mathcal{S}^{-1}(f)\right\}$$

$$= \inf\left\{\sum_{i\in I} \inf\left\{\|\mu_i\|_{\Psi_i}^{p} : \mu_i \in A_i^{-1}(f_i)\right\} : (f_i)_{i\in I} \in \mathcal{S}^{-1}(f)\right\}$$

for $f \in \sum_{i \in I}^p \mathcal{B}_i$. The norm on the RKBS $\mathcal{B}$ is given by

$$
\begin{aligned}
\|f\|_{\mathcal{B}}^p &= \inf\left\{ \|(\mu_i)_{i \in I}\|_{\bigoplus_{i \in I}^p \Psi_i}^p : (\mu_i)_{i \in I} \in A^{-1}(f) \right\} = \inf \left\|A^{-1}(f)\right\|_{\bigoplus_{i \in I}^p \Psi_i}^p \\
&= \inf \left\| \widetilde{(A_i)_{i \in I}}^{-1} \circ \mathcal{S}^{-1}(f) \right\|_{\bigoplus_{i \in I}^p \Psi_i}^p = \inf \left\| \bigcup_{(f_i)_{i \in I} \in \mathcal{S}^{-1}(f)} \widetilde{(A_i)_{i \in I}}^{-1}\left((f_i)_{i \in I}\right) \right\|_{\bigoplus_{i \in I}^p \Psi_i}^p \\
&= \inf \bigcup_{(f_i)_{i \in I} \in \mathcal{S}^{-1}(f)} \left\| \widetilde{(A_i)_{i \in I}}^{-1}\left((f_i)_{i \in I}\right) \right\|_{\bigoplus_{i \in I}^p \Psi_i}^p \\
&= \inf \left\{ \inf \left\| \widetilde{(A_i)_{i \in I}}^{-1}\left((f_i)_{i \in I}\right) \right\|_{\bigoplus_{i \in I}^p \Psi_i}^p : (f_i)_{i \in I} \in \mathcal{S}^{-1}(f) \right\} \\
&= \inf \left\{ \inf \left\{ \sum_{i \in I} \|\mu_i\|_{\Psi_i}^p : (\mu_i)_{i \in I} \in \widetilde{(A_i)_{i \in I}}^{-1}\left((f_i)_{i \in I}\right) \right\} : (f_i)_{i \in I} \in \mathcal{S}^{-1}(f) \right\}
\end{aligned}
$$

for $f \in \mathcal{B}$. If we denote the set $\left\{ \sum_{i \in I} \|\mu_i\|_{\Psi_i}^p : (\mu_i)_{i \in I} \in \widetilde{(A_i)_{i \in I}}^{-1}\left((f_i)_{i \in I}\right) \right\}$ by $\mathcal{C}$, then we conclude the proof by showing that:

$$
\sum_{i \in I} \inf \left\{ \|\mu_i\|_{\Psi_i}^p : \mu_i \in A_i^{-1}(f_i) \right\} = \inf \mathcal{C}
$$

for all $(f_i)_{i \in I} \in \mathcal{S}^{-1}(f)$. To show that $\sum_{i \in I} \inf \left\{ \|\mu_i\|_{\Psi_i}^p : \mu_i \in A_i^{-1}(f_i) \right\}$ is a lower bound for $\mathcal{C}$, we note that $(\mu_i)_{i \in I} \in \widetilde{(A_i)_{i \in I}}^{-1}\left((f_i)_{i \in I}\right)$ is equivalent to

$$
(\mu_i)_{i \in I} \in \bigoplus_{i \in I}^p \Psi_i \text{ and } \forall i \in I, A_i \mu_i = f_i. \tag{A.2}
$$

Let $(\nu_i)_{i \in I} \in \widetilde{(A_i)_{i \in I}}^{-1}\left((f_i)_{i \in I}\right)$.

Then, by the equation A.2, we have that $\inf \left\{ \|\mu_i\|_{\Psi_i}^p : \mu_i \in A_i^{-1}(f_i) \right\} \leq \|\nu_i\|_{\Psi_i}^p$ for all $i \in I$. Thus, we deduce that $\sum_{i \in I} \inf \left\{ \|\mu_i\|_{\Psi_i}^p : \mu_i \in A_i^{-1}(f_i) \right\} \leq \sum_{i \in I} \|\nu_i\|_{\Psi_i}^p$. Now, we have to show that $\sum_{i \in I} \inf \left\{ \|\mu_i\|_{\Psi_i}^p : \mu_i \in A_i^{-1}(f_i) \right\}$ is the greatest lower bound of $\mathcal{C}$. Let $\mathbf{c}$ be an any lower bound of the set $\mathcal{C}$. Since we already assumed that $(f_i)_{i \in I} \in \mathcal{S}^{-1}(f)$, the norm of $(f_i)_{i \in I}$ in $\bigoplus_{i \in I}^p \mathcal{B}_i$ is finite. That is, we know that $\sum_{i \in I} \inf \left\{ \|\mu_i\|_{\Psi_i}^p : \mu_i \in A_i^{-1}(f_i) \right\} = \sum_{i \in I} \|f_i\|_{\mathcal{B}_i} < \infty$. We denote the set $\left\{ i \in I : \inf \left\{ \|\mu_i\|_{\Psi_i}^p : \mu_i \in A_i^{-1}(f_i) \right\} \neq 0 \right\}$ by $H$. Then, for $i \in I \setminus H$, $\inf \left\{ \|\mu_i\|_{\Psi_i}^p : \mu_i \in A_i^{-1}(f_i) \right\} = 0$. Hence, there is a sequence $\{\nu_i^n\}_{n \in \mathbb{N}} \in A_i^{-1}(f_i)$, so that $\|\nu_i^n - 0\|_{\Psi_i}^p \to 0$ as $n \to \infty$. Furthermore, since $A_i^{-1}(f_i)$ is a translation of $\ker A_i$, by the equation 3.1, $A_i^{-1}(f_i)$ is a closed subset in $\Psi_i$. Therefore, we deduce that

$$
0 \in A_i^{-1}(f_i) \text{ for all } i \in I \setminus H \tag{A.3}
$$

For the case of $H$, note that $H$ is a countable subset of $I$. Accordingly, we may take a reordering bijection $g : \mathbb{N} \to H$. By simply using the definition of the infimum, for any $1 > \epsilon > 0$ and for any $g(n) \in H$, there is a $\nu_{g(n)} \in A_{g(n)}^{-1}(f_{g(n)})$ such that

$$
\inf \left\{ \|\mu_{g(n)}\|_{\Psi_{g(n)}}^p : \mu_{g(n)} \in A_{g(n)}^{-1}(f_{g(n)}) \right\} + \frac{1}{4} \cdot \frac{1}{2^n} \cdot \epsilon > \|\nu_{g(n)}\|_{\Psi_{g(n)}}^p. \tag{A.4}
$$

Combining the above results, we obtain the following:

$$\sum_{i \in I} \inf \left\{ \|\mu_i\|^p_{\Psi_i} : \mu_i \in A_i^{-1}(f_i) \right\} + \epsilon \tag{A.5}$$

$$> \sum_{i \in I} \inf \left\{ \|\mu_i\|^p_{\Psi_i} : \mu_i \in A_i^{-1}(f_i) \right\} + \sum_{n=1}^{\infty} \frac{1}{4} \cdot \frac{1}{2^n} \cdot \epsilon \tag{A.6}$$

$$= \sum_{i \in H} \inf \left\{ \|\mu_i\|^p_{\Psi_i} : \mu_i \in A_i^{-1}(f_i) \right\} + \sum_{n=1}^{\infty} \frac{1}{4} \cdot \frac{1}{2^n} \cdot \epsilon \tag{A.7}$$

$$= \sum_{n=1}^{\infty} \left( \inf \left\{ \|\mu_{g(n)}\|^p_{\Psi_{g(n)}} : \mu_{g(n)} \in A_{g(n)}^{-1}(f_{g(n)}) \right\} + \frac{1}{4} \cdot \frac{1}{2^n} \cdot \epsilon \right) \tag{A.8}$$

$$> \sum_{n=1}^{\infty} \|\nu_{g(n)}\|^p_{\Psi_{g(n)}} = \sum_{i \in H} \|\nu_i\|^p_{\Psi_i}. \tag{A.9}$$

Define $\xi_i = \begin{cases} \nu_i & \text{if } i \in H, \\ 0 & \text{if } i \in I \setminus H \end{cases}$. Then, by the equation A.3 and equation A.4, we know that for all $i \in I$, $\xi_i \in A_i^{-1}(f_i)$. In addition, from the inequalities in equation A.9, we also know that $\sum_{i \in I} \|\xi_i\|^p_{\Psi_i} \le \sum_{i \in I} \|f_i\|^p_{\mathcal{B}_i} + 1 < \infty$. Thus, by the equation A.2, we deduce that $(\xi_i)_{i \in I} \in \widetilde{(A_i)_{i \in I}}^{-1} ((f_i)_{i \in I})$ (i.e., $\sum_{i \in I} \|\xi_i\|^p_{\Psi_i} \in \mathcal{C}$). Finally, the following show that $\sum_{i \in I} \inf \left\{ \|\mu_i\|^p_{\Psi_i} : \mu_i \in A_i^{-1}(f_i) \right\}$ is the greatest lower bound of $\mathcal{C}$:

$$\sum_{i \in I} \inf \left\{ \|\mu_i\|^p_{\Psi_i} : \mu_i \in A_i^{-1}(f_i) \right\} + \epsilon > \sum_{i \in H} \|\nu_i\|^p_{\Psi_i} = \sum_{i \in I} \|\xi_i\|^p_{\Psi_i} \ge \mathbf{c} \quad \text{for all } 1 > \epsilon > 0,$$

where $\mathbf{c}$ is a lower bound of the set $\mathcal{C}$. $\qquad\square$

## A.5 PROOF OF PROPOSITION 4.3

*Proof.* Define a feature map $\psi_1 : \mathcal{X} \to \Psi_1^*$ by $\psi_1(x) = \psi_2(x) \circ \xi$ for $x \in \mathcal{X}$ and a linear map $A_1 : \Psi_1 \to \mathbb{R}^{\mathcal{X}}$ by $(A_1 \mu)(x) = \langle \psi_1(x), \mu \rangle$ for $x \in \mathcal{X}$ and $\mu \in \Psi_1$. Then, we deduce that $A_1 = A_2 \circ \xi$. Furthermore, $\mathcal{B}_1 = (\Psi_1, \psi_1, A_1)$ is an RKBS. Consider the map $\overline{\xi} : \Psi_1 / \ker A_2 \circ \xi \to \Psi_2 / \ker A_2$ defined by $\overline{\xi}([\mu]) = [\xi(\mu)]$ for $[\mu] \in \Psi_1 / \ker A_2 \circ \xi$. Since $\ker(A_2 \circ \xi) = \xi^{-1}(\ker A_2)$, $\overline{\xi}$ is a well-defined vector space monomorphism. The remaining proof for establishing surjectivity and isometry is straightforward. $\qquad\square$

## A.6 PROOF OF THEOREM 4.4

*Proof.* By the Definition 3.4, there is a map $\psi : X \to M(\Omega)^*$ defined by $\psi(x) = \Lambda^*(\iota(\sigma(x, \cdot)))$ for $x \in X$. And there is an RKBS map $A : M(\Omega) \to \mathbb{R}^X$ defined by $(A(\mu))(x) = \langle \psi(x), \mu \rangle$ for all $x \in X$ and $\mu \in M(\Omega)$ such that

$$\mathcal{F}_\sigma(\mathcal{X}, \Omega) \underset{\mathcal{B}}{\cong} M(\Omega) / \ker A.$$

Let $\Theta : \bigoplus_{i \in I}^1 L^1(\mu_i) \to M(\Omega)$ be the isometric isomorphism defined in equation 2.4. Define a map $\overline{\psi} : X \to \left( \bigoplus_{i \in I}^1 L^1(\mu_i) \right)^*$ by $\overline{\psi}(x) = \psi(x) \circ \Theta$. And consider a map $\overline{A} : \bigoplus_{i \in I}^1 L^1(\mu_i) \to \mathbb{R}^X$ defined by $\overline{A} = A \circ \Theta$. Then, by the Lemma 4.3, we have that

$$M(\Omega) / \ker A \underset{\mathcal{B}}{\cong} \bigoplus_{i \in I}^1 L^1(\mu_i) / \ker \overline{A}.$$

Now, let $\Phi_0 : \bigoplus_{i \in I}^\infty \left( L^1(\mu_i) \right)^* \to \left( \bigoplus_{i \in I}^1 L^1(\mu_i) \right)^*$ be the isometric isomorphism defined in equation 2.1. For each $i \in I$, if we define a map $\overline{\psi}_i : X \to \left( L^1(\mu_i) \right)^*$ by $\overline{\psi}_i(x) = p_i \left( \Phi_0^{-1} \left( \overline{\psi}(x) \right) \right)$

for $x \in X$ and define a map $\overline{A}_i : L^1(\mu_i) \to \mathbb{R}^X$ by $\left(\overline{A}_i(h)\right)(x) = \langle \overline{\psi}_i(x), h \rangle$ for $x \in X$ and $h \in L^1(\mu_i)$, then by the Proposition 4.2, we can deduce that

$$\bigoplus_{i \in I}^1 L^1(\mu_i)/\ker \overline{A} \underset{\mathcal{B}}{\cong} \sum_{i \in I}^1 \mathcal{B}_i,$$

where $\mathcal{B}_i = (L^1(\mu_i), \overline{\psi}_i, \overline{A}_i)$ for all $i \in I$. We want to show that $\mathcal{B}_i$ is indeed $\mathcal{L}_\sigma(\mu_i)$ for all $i \in I$. Suppose for each $i \in I$, $\Xi^i : L^\infty(\mu_i) \to \left(L^1(\mu_i)\right)^*$ is the isometric isomorphism introduced in equation 2.3. According to the Definition 3.6, it suffices to verify that $\overline{\psi}_i(x) = \Xi^i(\sigma(x, \cdot))$ for all $x \in X$ and $i \in I$. This condition is equivalent to $\overline{\psi}(x) = \Phi_0\left(\left(\Xi^i(\sigma(x, \cdot))\right)_{i \in I}\right)$ for all $x \in \mathcal{X}$. Hence, we want to prove the following holds: $(\Lambda^* (\iota(\sigma(x, \cdot))) \circ \Theta) ((f_i)_{i \in I}) = \Phi_0\left(\left(\Xi^i(\sigma(x, \cdot))\right)_{i \in I}\right) ((f_i)_{i \in I})$ for all $x \in \mathcal{X}$ and $(f_i)_{i \in I} \in \bigoplus_{i \in I}^1 L^1(\mu_i)$. First, for the left-hand side, we have:

$$(\Lambda^* (\iota(\sigma(x, \cdot))) \circ \Theta) ((f_i)_{i \in I}) = \left\langle \Lambda^* (\iota(\sigma(x, \cdot))), \mathcal{M}(K) \sum_{i \in I} \rho_i \right\rangle$$

$$= \left\langle \iota(\sigma(x, \cdot)) \circ \Lambda, \mathcal{M}(K) \sum_{i \in I} \rho_i \right\rangle = \sum_{i \in I} \langle \iota(\sigma(x, \cdot)) \circ \Lambda, \rho_i \rangle$$

$$= \sum_{i \in I} \langle \iota(\sigma(x, \cdot)), \Lambda(\rho_i) \rangle = \sum_{i \in I} \langle \Lambda(\rho_i), \sigma(x, \cdot) \rangle$$

$$= \sum_{i \in I} \int_\Omega \sigma(x, w) d\rho_i(w) = \sum_{i \in I} \int_\Omega \sigma(x, w) f_i(w) d\mu_i(w).$$

Next, for the right-hand side, we have:

$$\Phi_0\left(\left(\Xi^i(\sigma(x, \cdot))\right)_{i \in I}\right) ((f_i)_{i \in I}) = \sum_{i \in I} \left\langle \Xi^i(\sigma(x, \cdot)), f_i \right\rangle = \sum_{i \in I} \int_\Omega \sigma(x, w) f_i(w) d\mu_i(w).$$

$\square$

### A.7    PROOF OF PROPOSITION 5.1

*Proof.* As we noted in the Definition 3.6, we can easily show that for a given $\pi \in P(\Omega)$, $\mathcal{L}_\sigma^2(\pi) \subset \mathcal{L}_\sigma^1(\pi)$ and $\|f\|_{\mathcal{L}_\sigma^1(\pi)} \leq \|f\|_{\mathcal{L}_\sigma^2(\pi)}$ for all $f \in \mathcal{L}_\sigma^2(\pi)$. Let $\{\mu_i\}_{i \in I}$ be a maximal singular family containing $\{\mu_i\}_{i \in [n]}$. Consider the map $\iota : \bigoplus_{i \in [n]}^2 \mathcal{L}_\sigma^2(\mu_i) \to \bigoplus_{i \in I}^1 \mathcal{L}_\sigma^1(\mu_i)$ defined by $\iota(\mathbf{x})(i) = \begin{cases} \mathbf{x}(i), & \text{if } i \in [n] \\ 0, & \text{if } i \in I \setminus [n] \end{cases}$ for $\mathbf{x} \in \bigoplus_{i \in [n]}^2 \mathcal{L}_\sigma^2(\mu_i)$. Since $\iota(\mathbf{x})(i) = \mathbf{x}(i) \in \mathcal{L}_\sigma^2(\mu_i) \subset \mathcal{L}_\sigma^1(\mu_i)$ for all $i \in [n]$, we know that $\iota(\mathbf{x}) \in \prod_{i \in I} \mathcal{L}_\sigma^1(\mu_i)$. Furthermore, by the following inequalities $\sum_{i \in I} \|\iota(\mathbf{x})(i)\|_{\mathcal{L}_\sigma^1(\mu_i)} = \sum_{i=1}^n \|\mathbf{x}(i)\|_{\mathcal{L}_\sigma^1(\mu_i)} \leq \sum_{i=1}^n \|\mathbf{x}(i)\|_{\mathcal{L}_\sigma^2(\mu_i)} < \infty$, we deduce that $\iota(\mathbf{x}) \in \bigoplus_{i \in I}^1 \mathcal{L}_\sigma^1(\mu_i)$. Thus, $\iota$ is well-defined linear map.

By the Remark 3.8, we can define the RKBS linear map for the sum of RKBSs $\mathcal{S}_1 : \bigoplus_{i \in I}^1 \mathcal{L}_\sigma^1(\mu_i) \to \mathbb{R}^{\mathcal{X}}$ by $\mathcal{S}_1((f_i)_{i \in I})(x) = \sum_{i \in I} f_i(x)$ for $x \in \mathcal{X}$ and $(f_i)_{i \in I} \in \bigoplus_{i \in I}^1 \mathcal{L}_\sigma^1(\mu_i)$. And let $\mathcal{S}_2 : \bigoplus_{i \in [n]}^2 \mathcal{L}_\sigma^2(\mu_i) \to \mathbb{R}^{\mathcal{X}}$ be the RKBS linear map defined by $\mathcal{S}_2((f_i)_{i \in [n]})(x) = \sum_{i \in [n]} f_i(x)$ for $x \in \mathcal{X}$ and $(f_i)_{i \in [n]} \in \bigoplus_{i \in [n]}^2 \mathcal{L}_\sigma^2(\mu_i)$. Now, consider the map $\overline{\iota} : \bigoplus_{i \in [n]}^2 \mathcal{L}_\sigma^2(\mu_i)/\ker \mathcal{S}_2 \to \bigoplus_{i \in I}^1 \mathcal{L}_\sigma^1(\mu_i)/\ker \mathcal{S}_1$ defined by $\overline{\iota}([\mathbf{x}]) = [\iota(\mathbf{x})]$ for $\mathbf{x} \in \bigoplus_{i \in [n]}^2 \mathcal{L}_\sigma^2(\mu_i)$. From the following fact

$$\iota^{-1}(\ker \mathcal{S}_1) = \left\{ \mathbf{x} \in \bigoplus_{i \in [n]}^2 \mathcal{L}_\sigma^2(\mu_i) : \iota(\mathbf{x}) \in \ker \mathcal{S}_1 \right\}$$

$$= \left\{ \mathbf{x} \in \bigoplus_{i \in [n]}^2 \mathcal{L}_\sigma^2(\mu_i) : \sum_{i \in I} (\iota(\mathbf{x})(i))(x) = 0 \text{ for all } x \in \mathcal{X} \right\} = \ker \mathcal{S}_2,$$

we deduce that $\bar{\iota}$ is well-defined monomorphism. If we consider the map $\widetilde{\mathrm{id}} = \hat{\mathcal{S}}_1 \circ \bar{\iota} \circ \hat{\mathcal{S}}_2^{-1}$ : $\sum_{i \in [n]}^{2} \mathcal{L}_\sigma^2(\mu_i) \to \sum_{i \in I} \mathcal{L}_\sigma^1(\mu_i)$, then it is indeed the identity map. Thus, we have $\sum_{i \in [n]}^{2} \mathcal{L}_\sigma^2(\mu_i) \subset \sum_{i \in I} \mathcal{L}_\sigma^1(\mu_i)$. Furthermore, by the Remark 4.5, we know that $\sum_{i \in I} \mathcal{L}_\sigma^1(\mu_i) = \mathcal{F}_\sigma(\mathcal{X}, \Omega)$ as a set equality. $\qquad\square$

## A.8 PROOF OF PROPOSITION 5.2

*Proof.* For fixed $i \in [n]$, define $\iota : L^2(\Omega, \pi_i) \to L^2(\Omega \times [0,1], \pi_i \otimes \delta_{i/n})$ by $\iota(h)(w, r) = \begin{cases} h(w) & \text{if } r = \frac{i}{n}, \\ 0 & \text{otherwise} \end{cases}$ for $w \in \Omega$ and $r \in [0,1]$ where $\delta_{i/n}$ is the Dirac measure centred on $i/n$ in $([0,1], \mathcal{B}([0,1]))$. Then, $\iota(h)$ is measurable with respect to $(\Omega \times [0,1], \mathcal{B}(\Omega \times [0,1]))$ and $\int_{\Omega \times [0,1]} |\iota(h)(w, r)|^2 d\pi_i \otimes \delta_{i/n} < \infty$. Thus, $\iota$ is well-defined linear map.

Now, define $A : L^2(\Omega, \pi_i) \to \mathbb{R}^{\mathcal{X}}$ by $(Ah)(x) = \int_\Omega \sigma_i(x, w)h(w)d\pi_i$ for $h \in L^2(\Omega, \pi_i)$ and $x \in \mathcal{X}$ and $B : L^2(\Omega \times [0,1], \pi_i \otimes \delta_{i/n}) \to \mathbb{R}^{\mathcal{X}}$ by $(B\tilde{h})(x) = \int_{\Omega \times [0,1]} \sigma(x, w, r)\tilde{h}(w, r)d\pi_i \otimes \delta_{i/n}$ for $\tilde{h} \in L^2(\Omega \times [0,1], \pi_i \otimes \delta_{i/n})$ and $x \in \mathcal{X}$, which are the RKBS linear maps introduced in the Definition 3.6. Since we know that

$$\iota^{-1}(\ker B) = \left\{ h \in L^2(\Omega, \pi_i) : \iota(h) \in \ker B \right\} = \left\{ h \in L^2(\Omega, \pi_i) : B(\iota(h)) = 0 \right\}$$

$$= \left\{ h \in L^2(\Omega, \pi_i) : \int_\Omega \int_{[0,1]} \sigma(x, w, r)\iota(h)(w, r)d\delta_{i/n}d\pi_i \right\}$$

$$= \left\{ h \in L^2(\Omega, \pi_i) : \int_\Omega \sigma_i(x, w)h(w)d\pi_i \right\} = \ker A,$$

it follows that $\bar{\iota} : L^2(\Omega, \pi_i) \to L^2(\Omega \times [0,1], \pi_i \otimes \delta_{i/n})$ defined by $\bar{\iota}([h]) = [\iota(h)]$ for $h \in L^2(\Omega, \pi_i)$ is well-defined monomorphism.

Consider a map $\tilde{\mathrm{id}} = \hat{B} \circ \bar{\iota} \circ \hat{A}^{-1} : \mathcal{L}_{\sigma_i}^2(\Omega, \pi_i) \to \mathcal{L}_\sigma^2(\Omega \times [0,1], \pi_i \otimes \delta_{i/n})$ and let $Ah \in \mathcal{L}_{\sigma_i}^2(\Omega, \pi_i) = \mathrm{im}(A)$. Then, we have $\tilde{\mathrm{id}}(Ah) = \tilde{B} \circ \bar{\iota}([h]) = \hat{B}([\iota(h)]) = B\iota(h) = Ah$. It means that $\tilde{\mathrm{id}}$ is an identity map. Furthermore, we can deduce that

$$\|Ah\|_{\mathcal{L}_\sigma^2(\Omega \times [0,1], \pi_i \otimes \delta_{i/n})} = \|B\iota(h)\|_{\mathcal{L}_\sigma^2(\Omega \times [0,1], \pi_i \otimes \delta_{i/n})} = \|[\iota(h)]\|_{L^2(\Omega \times [0,1], \pi_i \otimes \delta_{i/n})/\ker B}$$

$$= \inf_{\tilde{g} \in \ker B} \|\iota(h) + \tilde{g}\|_{L_\sigma^2(\Omega \times [0,1], \pi_i \otimes \delta_{i/n})} \leq \inf_{g \in \ker A} \|\iota(h) + \iota(g)\|_{L_\sigma^2(\Omega \times [0,1], \pi_i \otimes \delta_{i/n})}$$

$$= \|Ah\|_{\mathcal{L}_{\sigma_i}^2(\Omega, \pi_i)}.$$

Thus, for $i = 1, \ldots, n$, we have

$$\mathcal{L}_{\sigma_i}^2(\Omega, \pi_i) \subset \mathcal{L}_\sigma^2(\Omega \times [0,1], \pi_i \otimes \delta_{i/n})$$

and for all $f \in \mathcal{L}_\sigma^2(\Omega, \pi_i)$, $\|f\|_{\mathcal{L}_\sigma^2(\Omega \times [0,1], \pi_i \otimes \delta_{i/n})} \leq \|f\|_{\mathcal{L}_{\sigma_i}^2(\Omega, \pi_i)}$. From this, we can verify that $\sum_{i \in [n]}^{2} \mathcal{L}_{\sigma_i}^2(\Omega, \pi_i) \subset \sum_{i \in [n]}^{2} \mathcal{L}_\sigma^2(\Omega \times [0,1], \pi_i \otimes \delta_{i/n})$ and since $\{\pi_i \otimes \delta_{i/n}\}_{i=1}^{n}$ is a singular family in $P(\Omega \times [0,1])$, by the Proposition 5.1, we conclude that $\sum_{i \in [n]}^{2} \mathcal{L}_{\sigma_i}^2(\Omega, \pi_i) \subset \mathcal{F}_\sigma(\mathcal{X}, \Omega \times [0,1])$. $\qquad\square$

