# OpenReview forum: "Decomposition of one-layer neural networks via the infinite sum of reproducing kernel Banach spaces"
_ICLR.cc/2025/Conference — Submitted to ICLR 2025_

### Official Review · Reviewer_mrzL · 2024-10-28

**Soundness:** 3
**Presentation:** 2
**Contribution:** 3
**Rating:** 6
**Confidence:** 3

**Summary:**

Reproducing kernel Hilbert spaces (RKHS) can be decomposed into a sum of RKHS. A natural generalization is to consider reproducing kernel _Banach_ spaces (RKBS) and to decompose it as a sum of RKBSs. Defining the sum is non-trivial, and the authors take on this task. Furthermore, given an RKBS with an integral the authors provide a way to decompose it into a sum of RKBSs. The authors claim that this stablishes a connection to neural networks.

**Strengths:**

1. Banach spaces have a much wider range of options than Hilbert spaces, which seems like a good motivation to consider this type of spaces.
2. The authors hint to novel ideas connecting integral RKBS to the study of neural networks (however, see Weakness 1).
3. The authors are very comprehensive to a reader – such as myself, who has not seen several of the relevant concepts since a real analysis course.

**Weaknesses:**

1. The authors claim that there is a connection with neural networks, but do not make it clear nor precise. For example, the only mention of neural networks are in the introduction and a single mention in Subsection 3.3, without going into detail of the correspondence between the terms developed in the paper and neural networks.
2. Immediately after Proposition 3.7 the authors mention the feature map ($s$) and the RKBS ($\mathcal{S}$), without an explicit definition in the main text. As the definitions are available in the appendix, I think it would strengthen the paper to include them in the main text.
3. Presumably there is a correspondence between the triples $(\Psi,\psi,A)$ that the authors see with neural network concepts, but with the current status of the paper is quite hard to understand. Can the authors make this relationship explicit?

**Questions:**

### Questions:
1. How important is the compactness of $\Omega$ for the main results? By the work of Neal (1996) we know that for _decent_ densities (e.g., finite moments and bounded activation functions) we have a kernel similar to the kernel of $L^2$ stated in line 299, even when $\Omega$ is unbounded.
2. It is not immediately clear to me why $\mathrm{Im}(A)$ has finite dimension as indicated in line 284. Perhaps I am missing something. Is there a specific reference or lemma that makes it clear?
3. The comment on lines 414-417, about the Tietze extension, seems out of place. Should it be in the proof of Proposition 5.2?

### Citation issues:
- Citations are not consistent throughout, which makes it harder on the reader to digest this technical paper. A good guide is in https://guides.library.unr.edu/apacitation/in-textcite. An example of this is the third line of the first section, which includes the name of the authors twice.
- Proper names are typically capitalized, even in the references. e.g. 'Banach' in Bartolucci et al (2023).

### Minor notation issues:
- The authors use $\langle \cdot ,\cdot \rangle$  and $<\cdot ,\cdot>$ interchangeably for inner products. It would be good to standardize notation, or clarify what is the main difference between these two notations. For example, in page 6, line 291 uses $<\cdot ,\cdot>$ while line 320 uses $\langle \cdot ,\cdot \rangle$ , with no clear difference between them. Perhaps one of the two notations corresponds to a semi-inner product, but at the moment this distinction is not at all clear.
- I could not find a definition of $\mu \perp\nu$ in the text, used in line 161. I assume it means something like $\int_{\Omega} \mu(\omega)d\nu(\omega)=0$, or something in that sense. This could be easily added.

### Minor grammatical mistakes:
- Line 350, 'an another' should just be 'another'
- There is a repetition of "equation" in Remark 3.8.

### References

Francesca Bartolucci, Ernesto De Vito, Lorenzo Rosasco, and Stefano Vigogna. Understanding neural networks with reproducing kernel Banach spaces. Applied and Computational Harmonic Analysis, 62:194–236, 2023

Radford M Neal. Priors for infinite networks. In Bayesian Learning for Neural Networks, Lecture Notes in  Statistics, pp. 29–53. Springer New York, New York, NY, 1996. ISBN 0387947248.

---

> ### Author Response · Authors · 2024-11-18
>
> Thank you for your insightful observations and attention to detail. The points you raised are excellent questions that not only prompt deep reflection for us but also for other readers. Moreover, many of the aspects you highlighted have been immensely helpful in allowing us to refine and further develop the content of our paper.
>
> Weakness1. The authors claim that there is a connection with neural networks, but do not make it clear nor precise. For example, the only mention of neural networks are in the introduction and a single mention in Subsection 3.3, without going into detail of the correspondence between the terms developed in the paper and neural networks.
>
> Answer:
> It seems we did not elaborate enough on the relationship between neural networks and integral RKBSs. As you are well aware, in the ERM framework, we evaluate the performance of machine learning models by setting the hypothesis space that the model can represent and conducting various analyses within this space. As mentioned in the introduction, we consider integral RKBSs as hypothesis spaces that neural networks are believed to generate (or represent). To ensure a clearer understanding, we will add more detailed descriptions of the relationship between neural networks and integral RKBSs and notify you accordingly. Thank you.
>
> Weakness2.
> Immediately after Proposition 3.7 the authors mention the feature map ($\mathbf{s}$) and the RKBS map ($\mathcal{S}$), without an explicit definition in the main text. As the definitions are available in the appendix, I think it would strengthen the paper to include them in the main text.
>
> Answer: Thank you for your insightful comment. We had not considered this, and it is certainly a point that could confuse many readers. In order to continue our discussion, we should have considered the RKBS triple for the sum defined in Proposition 3.7, but we made a mistake in clearly stating this. We will correct it and upload the revision.
>
> Weakness3.
> Presumably there is a correspondence between the triples $(\Psi,\psi,A)$ that the authors see with neural network concepts, but with the current status of the paper is quite hard to understand. Can the authors make this relationship explicit?
>
> Answer:The RKBS triple $(\Psi, \psi, A)$ concept we use is analogous to what is suggested in traditional kernel methods (Hilbert space). To provide an intuitive explanation, through the map $\psi: \mathcal{X} \rightarrow \Psi^{\*},$ we can view the information (points) from the data space $\mathcal{X}$ as information in the high-dimensional abstract space $\Psi^{*}$. This enables us to distinguish features of the data that could not be separated in the original data space $\mathcal{X}$, but can be in $\Psi^{\*}$. The linear map $A$ serves to transform the information, now distinguishable in the abstract space, into a function space (RKBS) that we can learn. To help with further understanding, we would like to explain why the hypothesis space for neural networks is made into an RKBS in relation to Weaknesses 1 and 3.
>
> The first reason for using RKBS and RKHS as hypothesis spaces in machine learning is that we expect our hypothesis space (function space) to satisfy at least completeness and pointwise continuity. This is because, when we aim to find a target function through a machine learning model, the approximation process of the target function relies on the metric (or topology) of the hypothesis space, and this approximation process must converge at least pointwise (the minimal assumption that two functions in the function space are close).
>
> The second reason is that, in order to obtain the existence of a solution to the problem we are trying to solve through machine learning, we need to demonstrate the Representer theorem. In many cases, the Representer theorem is derived under the assumption of RKHS and RKBS. For this reason, in paper [1], the hypothesis space for neural networks is defined as the integral RKBS, and the Representer theorem is shown. However, demonstrating such existence does not directly lead to a concrete algorithm, which we believe is one of the main reasons why deep learning is often referred to as a black box.
>
> [1] Francesca Bartolucci et al. “Understanding neural networks with reproducing kernel Banach spaces”. In: Applied and Computational Harmonic Analysis 62 (2023), pp. 194–236.

---

> > ### Author Response · Authors · 2024-11-18
> >
> > Question1.
> > How important is the compactness of $\Omega$ for the main results? By the work of Neal (1996) we know that for decent densities (e.g., finite moments and bounded activation functions) we have a kernel similar to the kernel of $L^2$ stated in line 299, even when $\Omega$ is unbounded.
> >
> > Answer:This is a thought-provoking question that gets to the core of the paper. Basically, we introduced the compactness of $\Omega$ for the convenience of the argument. Due to time constraints, we couldn't locate the relevant content in the reference you mentioned, but the assumptions of finite moments and bounded activation functions are precisely the properties we aim to obtain by assuming the compactness of $\Omega$. In line 299, we cite the results of [1]. In Appendix A of this paper, it is shown that the existence of the kernel of $\mathcal{L}^{2}\_{\sigma}(\pi)$ requires the assumption of compactness, and we intended to reference this result reliably. Finally, as we mention in Remark 4.5, by assuming compactness, we made the space $\mathcal{L}^{1}\_{\sigma}(\mu\_{i})$ separable which opens the way for further research. This is because, as can be seen in Theorem 2.4 of [2], separable RKBSs admit reproducing kernels.
> >
> > Question2.
> > It is not immediately clear to me why $\operatorname{Im}(A)$ has finite dimension as indicated in line 284. Perhaps I am missing something. Is there a specific reference or lemma that makes it clear?
> >
> > Answer: This follows from the properties of compact operators. Given two Banach spaces $X$ and $Y$, if $T: X \rightarrow Y$ is a bounded compact operator and the image of $T$ is closed in $Y$, then the image of $T$ must be finite-dimensional. A specific reference for this result is given in Chapter 4, Section 3, Exercise 4 in [3]
> >
> > Question3.
> > The comment on lines 414-417, about the Tietze extension, seems out of place. Should it be in the proof of Proposition 5.2?
> >
> > Answer:  Yes, it is needed to construct a specific continuous function $\sigma$. As seen in lines 828–834, we used the fact that $\sigma(x,w,\frac{i}{n}) = \sigma_{i}(x,w)$. Furthermore, in order to reliably define the integral RKBS $\mathcal{F}\_{\sigma}(\mathcal{X}, \Omega)$, we used it to make $\sigma$ a continuous function defined on $\mathcal{X} \times \Omega \times [0,1]$.
> >
> > Regarding citation issues, minor notation issues, and minor grammatical mistakes:
> > Thank you so much for kindly pointing out our shortcomings. It seems like a great learning opportunity for us. We will correct it and upload the revision. Once again, thank you
> >
> >
> > [1] Francis Bach. “Breaking the curse of dimensionality with convex neural networks”. In: The Journal of Machine Learning Research 18.1 (2017), pp. 629–681.
> >
> > [2] Rong Rong Lin, Hai Zhang Zhang, and Jun Zhang. “On reproducing kernel Banach spaces: Generic definitions and unified framework of constructions”. In: Acta Mathematica Sinica, English Series 38.8 (2022), pp. 1459–1483.
> >
> > [3] Conway, J. B. (1997). A Course in Functional Analysis (2nd ed.). Graduate Texts in Mathematics, 96. Springer-Verlag. ISBN 978-0387972459

---

> ### Author Response · Authors · 2024-11-23
>
> Dear Reviewer mrzL,
>
> Thank you very much for your valuable feedback. In response to Weakness1 you raised, we have added an explanation regarding the relationship with one-layer neural networks in the revised Section 3.3. Additionally, as per your suggestion, we have further elaborated on Weakness2 in Proposition 3.7 and have uploaded the updated version. We have also addressed the citation and minor notation issues you pointed out.
>
> We truly appreciate the time and effort you took in reviewing our paper, and your feedback has been incredibly helpful in improving our work. Once again, thank you.

---

> > ### Comment · Reviewer_mrzL · 2024-11-25
> >
> > The update looks good. I think the connection to neural networks can be strengthened even further. Nevertheless, with the current state I can see the point of the authors, which is sufficient to raise my score to 6. I am still quite intrigued by the fact that you need compactness of $\Omega$ for the result to hold.
> >
> > Regarding the result of Neal (1996), he proves that when considering a Bayesian neural network under decent conditions (finite variance of priors, and bounded activation functions), the limit of the standardized infinite width regime converges to a Gaussian process (GP) with a known covariance function, by a simple application of CLT. The connection to GPs allows for a clear connection with the L2 space, and its corresponding reproducing kernel. Further results have obtained explicit covariances (e.g., Cho and Saul 2009, Lee et al 2018, de Matthews et al 2018, Yang 2020).
> >
> > All of that to say that I strongly suspect that the compactness assumption could be dropped in one way or the other.
> >
> > ##### References:
> >
> > Youngmin Cho and Lawrence Saul. Kernel methods for deep learning. In Y. Bengio, D. Schuurmans, J. Lafferty, C. Williams, and A. Culotta (eds.), Advances in Neural Information Processing Systems, volume 22. Curran Associates, Inc., 2009. URL https://proceedings.neurips.cc/paper_files/paper/2009/file/5751ec3e9a4feab575962e78e006250d-Paper.pdf
> >
> > Alexander G. de G. Matthews, Jiri Hron, Mark Rowland, Richard E. Turner, and Zoubin Ghahramani. Gaussian process behaviour in wide deep neural networks. In International Conference on Learning Representations, 2018. URL https://openreview.net/forum?id=H1-nGgWC-.
> >
> > Jaehoon Lee, Jascha Sohl-dickstein, Jeffrey Pennington, Roman Novak, Sam Schoenholz, and Yasaman Bahri. Deep neural networks as Gaussian processes. In International Conference on Learning Representations, 2018. URL https://openreview.net/forum?id=B1EA-M-0Z
> >
> > Radford M Neal. Priors for infinite networks. In Bayesian Learning for Neural Networks, Lecture Notes in  Statistics, pp. 29–53. Springer New York, New York, NY, 1996. ISBN 0387947248.
> >
> > Greg Yang. Tensor programs I: wide feedforward or  recurrent neural networks of any architecture are Gaussian processes. In 33rd Conference on Neural Information Processing Systems (NeurIPS 2019), 2019. URL http://arxiv.org/abs/1910.12478

---

> > > ### Author Response · Authors · 2024-11-26
> > >
> > > We sincerely appreciate your valuable response.
> > >
> > > We also strongly agree with your perspective and are hopeful about removing the compactness assumption. Likewise, we positively view the direction of extending the parameter space
> > > $\Omega \subset_{cpt} \mathbb{R}^{D}$ to $\mathbb{R}^{D}$. We apologize for not fully explaining our reasoning in response to Question 1, and we would like to make a few additional comments on this matter.
> > > As mentioned above, please understand that the compactness assumption was primarily introduced for simplicity in our arguments.
> > >
> > > 1. Regarding the parts of our paper where the compactness assumption is necessary (Proposition 3.5):
> > >
> > > In fact, Proposition 3.5 is somewhat tangential to the main context of our paper. This theorem was included to aid understanding of the class of integral RKBS, as its structure diverges from the intuition we typically derive from the Universal Approximation Theorem.
> > > In the proof of Proposition 3.5, we applied Exercise 8.10.134 in reference [1], which implicitly relies on the fact that any measure on a compact space is tight. Therefore, it is unclear whether the same argument can be applied in the general case of $\mathbb{R}^{D}$.
> > > (Additionally, we would like to suggest considering the interesting result in [2], which shows that no RKHS contains the space of continuous functions defined on a compact metric space.)
> > >
> > > 2. Regarding a purely mathematical perspective:
> > >
> > > If we aim to extend the parameter space
> > > $\Omega$ beyond a subset of $\mathbb{R}^{D}$ to a general topological space $(\Omega,\tau)$ (not restricted to Euclidean spaces), compactness certainly presents some advantages. Specifically, any Borel measure defined on a compact metric space is a Radon measure. In our arguments, we indeed had to handle the space of Radon measures, as we relied on the Riesz Representation Theorem. However, by assuming compact metric space, we were able to work with the more tractable space of Borel measures instead. Of course, as you pointed out, when the parameter space is extended to all of $\mathbb{R}^{D}$, the space remains complete and separable, so the same argument can still apply (see Theorem 7.1.7 in [1]).
> > >
> > >
> > > We greatly appreciate your insightful feedback. We also believe that a significant portion of our research can be developed on $\mathbb{R}^{D}$. Furthermore, we are grateful for the abundant references you provided regarding NNGP (Neural Network Gaussian Process). In particular, we have learned a great deal about the statistical perspective from the reference [3] you provided. Once again, thank you very much.
> > >
> > > References:
> > >
> > >
> > > [1] Bogachev, Vladimir Igorevich, and Maria Aparecida Soares Ruas. Measure theory. Vol. 2. Berlin: springer, (2007).
> > >
> > >
> > > [2] Steinwart, Ingo. "Reproducing kernel Hilbert spaces cannot contain all continuous functions on a compact metric space." Archiv der Mathematik 122.5 (2024): 553-557.
> > >
> > >
> > > [3]Radford M Neal. Priors for infinite networks. In Bayesian Learning for Neural Networks, Lecture Notes in Statistics, pp. 29–53. Springer New York, New York, NY, 1996. ISBN 0387947248.

---

### Official Review · Reviewer_Ubdx · 2024-11-02

**Soundness:** 3
**Presentation:** 3
**Contribution:** 3
**Rating:** 8
**Confidence:** 3

**Summary:**

The authors represent neural networks using an integral RKBS, where the feature space is the measure space corresponding to the distribution of the weight of the final layer. They characterize the decomposition of RKBSs via the decompsition of the feature spaces, and show the integral RKBS representing neural networks is deomposed into the sum of a family of p-norm RKBSs, each of which is characterized by a probability measure.

**Strengths:**

Applying the theory of RKBSs to analyzing neural networks and reducing the problems to those in the feature spaces of the RKBSs is interesting approach. The result is solid and the mathematical notions are carefully introduced. I think this paper provides a direction of future studies of the theory of neural networks.

**Weaknesses:**

Although this paper is well-organized and the mathematical notions are clear, for readers in the machine learning community, I think more explanations that shows the connection between the theoretical approaches and results and the pratical neural networks.
  - In my understanding, the decomposition is by virtue of the decomposition of the measure space (feature space), and that is why RKBSs are useful in the analysis of neural networks. I think the reason why RKBSs are useful should be clearly explained in the main text.
  - The motivation of the decomposition should be explained from the perspective of neural networks. I thought that since the pratical neural networks are represented by the sum involving the weight, instead of the integral involving the distribution of the weight, the decomposition of the integral RKBS into the sum of the family of smaller RKBSs makes the representation more practical. Does this interpretation correct? I think the advavtage of the decomposition should be discussed from the perspective of the application to the analysis of neural networks.
  - Maybe related to the above point, but can we construct the family $\{\mu_i\}$ in Theorem 4.4 explicitly? I think understanding $\{\mu_i\}$ is important for the analysis of the weights of neural networks. Do you have any examples or any comments on this point?

If the motivation related to neural networks becomes clearer, I will consider raising my score.

**Questions:**

Questions:
- In definition 3.4, $\sigma$ can be any element in $C(\mathcal{X}\times\Omega)$. Does this mean we can deal with *deep* neural networks by properly setting $\sigma$? In that case, I think this framework is more flexible than other methods like ridgelet transform [1] (in the framework of ridgelet transform, we have to consider the form $\sigma(\langle w,x\rangle-b)$). Can we apply this framework of RKBS to show the universality of deep neural networks?

Minor comments:
- p5, line 276, "when $\mathcal{X}$ be ..." should be "when $\mathcal{X}$ is ..." ?
- In Remark 3.8, "equation equation 3.2" should be "equation 3.2".

References:
[1] S. Sonoda and N. Murata, "Neural network with unbounded activation functions is universal approximator", Applied and Computational Harmonic Analysis, 43(2): 233-268.

---

> ### Author Response · Authors · 2024-11-18
>
> Thank you for your positive evaluation of our research results. Your insightful questions are closely related to the future research directions we have in mind. We appreciate the opportunity to elaborate on this. Additionally, Question 1 was a profound question that we had not anticipated. We will do our best to provide thorough and sincere answers to all of these points. I would like to express my gratitude for taking the time to review.
>
> Weakness1.
> In my understanding, the decomposition is by virtue of the decomposition of the measure space (feature space), and that is why RKBSs are useful in the analysis of neural networks. I think the reason why RKBSs are useful should be clearly explained in the main text.
>
> Answer:
> Your insights and feedback are excellent. We apologize for the insufficient explanation regarding the theoretical development using RKBS. We will provide a separate explanation on the reasons for using RKBS and its usefulness.
>
> 1.The reason for using RKBS as the hypothesis space for neural networks is as follows. When defining a hypothesis space (function space) in machine learning, we consider completeness and pointwise convergence as the minimal assumptions required for the properties of the function space. This is because, when approximating a target function using the metric (or topology) of the hypothesis space in machine learning (i.e., when trying to approximate it through the learning algorithm), the function approximated by the learning algorithm must pointwise converge to the target function.
>
> 2.Advantages of using RKBS: The useful property of RKBS (characterization theorem (Theorem 3.3)) allows us to always consider RKBS as a triple of feature space, feature map, and RKBS map (i.e., $(\Psi, \psi, A)$ in our submission), which enables us to handle the feature space (=measure space) of neural networks. This plays a crucial role in leading the decomposition of the feature space (=measure space) of neural networks (in the Banach space sense) to the decomposition of the hypothesis space of neural networks (in the RKBS sense), as shown in our compatibility theorem (Proposition 4.2). We will elaborate on this point and provide the revision.
>
> Weakness2.
> The motivation of the decomposition should be explained from the perspective of neural networks. I thought that since the pratical neural networks are represented by the sum involving the weight, instead of the integral involving the distribution of the weight, the decomposition of the integral RKBS into the sum of the family of smaller RKBSs makes the representation more practical. Does this interpretation correct? I think the advavtage of the decomposition should be discussed from the perspective of the application to the analysis of neural networks.
>
> Answer: Yes, that’s correct. Your interpretation aligns with one of our motivations for decomposing the integral RKBS.
> To begin with, in order to ensure the existence of a solution to the problem we aim to solve through machine learning, the Representer theorem is essential. One paper that successfully proves the Representer theorem for neural networks is Theorem 3.9. in [1]. However, it is important to note that guaranteeing the existence of a solution and obtaining a concrete algorithm are entirely different matters. To be more specific, there is no direct method to find the extremal points in Theorem 3.9 above. Therefore, it is likely that the optimization algorithms currently used with neural networks do not guarantee the existence of a solution as stated in the Representer theorem. In an effort to address this issue, we decomposed the hypothesis space of neural networks, the integral RKBS $\mathcal{F}\_{\sigma}(\mathcal{X},\Omega)$, into more tractable components, such as $\mathcal{L}\_{\sigma}^{1}(\mu\_{i})$, while preserving the RKBS structure. In our approach, the $\mathcal{L}\_{\sigma}^{1}(\mu\_{i})$ spaces are separable, and by the Theorem 2.4. in [2], we can also ensure the existence of the kernel. We thought that if the mathematical formulation was not fully refined, it could lead to misunderstandings by the readers. Additionally, since we were not confident enough, we decided not to mention our expectations in the manuscript. We believe that the comment on Weakness 3 could help us explain our perspective on the applications more intuitively, so we will continue to elaborate on this point in the Weakness 3.
>
> [1] Francesca Bartolucci et al. “Understanding neural networks with reproducing kernel Banach spaces”. In: Applied and Computational Harmonic Analysis 62 (2023), pp. 194–236.
>
> [2] Rong Rong Lin, Hai Zhang Zhang, and Jun Zhang. “On reproducing kernel Banach spaces: Generic definitions and unified framework of constructions”. In: Acta Mathematica Sinica, English Series 38.8 (2022), pp. 1459–1483.

---

> ### Author Response · Authors · 2024-11-18
>
> Weakness3.
> Maybe related to the above point, but can we construct the family $\mu\_{i}$ in Theorem 4.4 explicitly? I think understanding $\mu\_{i}$ is important for the analysis of the weights of neural networks. Do you have any examples or any comments on this point?
>
> Answer:
> This is an important question from an application perspective. We also believe that understanding $\mu\_{i}$, which corresponds to the weights of neural networks, is crucial for our analysis. However, the family of $\\{\mu\_{i}\\}\_{i\in I}$  we use in Theorem 4.4 are constructed via Zorn's Lemma as described in lines 161-162, so they cannot be explicitly computed. In this regard, we have introduced Proposition 5.1, Proposition 5.2, and Remark 5.3 as part of a discussion on future research directions for applications. Let me explain Proposition 5.1. As you know, $\mathcal{L}\_{\sigma}^{2}(\pi)$ is a model for one-layer neural networks with infinite width, where the input-layer parameters are fixed by the distribution $\pi$, and only the layer-output parameters are learned. This model is formulated in the framework of RKHS (Please see [1], [2]). We consider an arbitrary finite singular probability measure $\\{\mu\_{i}\\}\_{i \in [n]}$ defined on the input-layer parameter space $\Omega$, and (intuitively, if we decompose the parameter space $\Omega$ into a finite number of domains), we showed that the sum space $\sum\_{i\in[n]}^{2}\mathcal{L}\_{\sigma}^{2}(\mu\_{i})$ of the RKHSs derived from this measure family is embedded into the integral RKBS $\mathcal{F}\_{\sigma}(\mathcal{X},\Omega)$, which serves as the hypothesis space for one-layer neural networks. This implies that the multiple kernel methods for the models discussed earlier perform worse in approximation power than the one-layer neural networks. Furthermore, our theory extends beyond the case where $p \neq 2$ and includes infinite (countably infinite singular measure families). The diagram below illustrates this situation.
> $$
> \\{\text{ family of singular probability measures } \mu\_{i} \text{ on } \Omega\\} \rightarrow \sum\_{i\in [n]}^{p}\mathcal{L}\_{\sigma}^{p}(\mu\_{i}) \overset{\text{distance}}{\hookrightarrow} \mathcal{F}\_{\sigma}(\mathcal{X},\Omega) = \sum\_{i\in I}^{1}\mathcal{L}\_{\sigma}^{1}(\mu\_{i})
> $$
> As mentioned earlier in Weakness 2, the current method for optimizing neural networks is not one that guarantees the existence of solutions based on the Representer theorem. We consider a bottom-up approach to asymptotically approximate the integral RKBS $\mathcal{F}\_{\sigma}(\mathcal{X},\Omega)$ as our future research direction, and we aim to use this approach to find specific algorithms that ensure the existence of solutions as guaranteed by the Representer theorem.
>
>
> [1] Francis Bach. “Breaking the curse of dimensionality with convex neural networks”. In: The Journal of Machine Learning Research 18.1 (2017), pp. 629–681.
>
> [2] Ali Rahimi and Benjamin Recht. “Random features for large-scale kernel machines”. In: Advances in neural information processing systems 20 (2007).

---

> ### Author Response · Authors · 2024-11-18
>
> Question1.
> In definition 3.4, $\sigma$ can be any element in $C(\mathcal{X}\times\Omega)$. Does this mean we can deal with deep neural networks by properly setting $\sigma$? In that case, I think this framework is more flexible than other methods like ridgelet transform [1] (in the framework of ridgelet transform, we have to consider the form $\sigma(\left<w,x\right>-b)$). Can we apply this framework of RKBS to show the universality of deep neural networks?
>
> Answer:
> That is truly an insightful question! You've raised a great point regarding the notation we used. Additionally, the references you cited are not only excellent in content but also serve as a great example that highlights the core of the question. To clarify, let me explain in a few steps.
>
> 1.Regarding the flexibility of $\sigma$: As you pointed out, defining $\sigma(x, w) = g(\left<x, \hat{w}\right> - b)$ in our setting precisely models a one-layer neural network. In the original paper [2] where integral RKBS was first defined, the method you mentioned was used exactly as you described. However, there are several reasons for using a more flexible notation like the one we adopted.
> First, it was used to clearly establish that $\sigma(x, \cdot) \in C(\Omega)$ for all $x \in \mathcal{X}$.
> Second, despite this flexibility, it did not hinder the proof of our results in any way.
> Third, as you mentioned, we have not excluded the possibility that deep neural networks themselves can be represented by integral RKBS, and this will be further elaborated later.
> Finally, let me explain the advantages of using such a highly flexible representation. Our Proposition 5.2 shows that the hypothesis space formed by summing RKHSs, which are constructed by setting input-layer parameters from different activations and distributions, can be embedded into the integral RKBS with the activation function we have constructed. This is likely a result that cannot be obtained using a more limited notation such as $g(\left<x, \hat{w}\right> - b)$. We believe this result indirectly demonstrates how increasing the dimensionality of parameters, such as the bias term, can significantly enhance the expressive power of the model.
>
> 2. Regarding the referenced paper [1]: While it may not be directly related to your question, I would like to provide additional explanation for clarity. Although I have not had time to read it in detail, I will do my best to explain based on what I know. First, the notation in question seems to refer to equation (2) in the paper:
> $$\int_{\mathbb{Y}^{m+1}}T(\mathbf{a},b)\eta(\mathbf{a}\cdot \mathbf{x}-b)d\mu(\mathbf{a},b)$$
> which is similar to the form expressed by the elements of $\mathcal{L}\_{\sigma}^{p}(\mu)$
> in our submission (though the notion of metric between elements differs since ours is formulated as an RKBS).
> The cited paper demonstrates the
> $L^{2}$-sense universality of shallow neural networks defined on an unbounded parameter space, $\mathbb{Y}^{m+1} = \mathbb{R}^{m+1}$(Theorem 5.11). To efficiently handle unbounded activations defined on an unbounded parameter space, the authors employed elegant concepts from distribution theory in their proof. However, universal approximation itself is somewhat different from our objective. Our focus lies in analyzing the hypothesis space of neural networks. By working with a compact parameter space $\Omega$, we avoided challenges associated with unbounded activations.
>
> 3.Regarding universality: As you may know, the universality of neural networks can be proven in various ways depending on the objective. While there are multiple classification schemes, I would like to focus on approaches that explicitly utilize the hypothesis space of neural networks. Specifically, Theorem 3.8 of [3] provides an example of this methodology. One advantage of such an approach, compared to traditional studies on universality, is that it allows us to directly control the approximation quality by using the norm of the hypothesis space, which is the same concept of distance used during actual machine learning training. The paper we referenced, focuses on Barron spaces. However, Theorem 2.3 of the same paper([3]) shows that when using the ReLU activation function, the Barron space and the integral RKBS are equivalent. Therefore, we believe the same results can be applied to the integral RKBS class.
>
> [1] S. Sonoda and N. Murata, "Neural network with unbounded activation functions is universal approximator", Applied and Computational Harmonic Analysis, 43(2): 233-268.
>
> [2] Francesca Bartolucci et al. “Understanding neural networks with reproducing kernel Banach spaces”. In: Applied and Computational Harmonic Analysis 62 (2023), pp. 194–236.
>
> [3]E Weinan and Stephan Wojtowytsch. “Representation formulas and pointwise properties for Barron functions”. In: Calculus of Variations and Partial Differential Equations 61.2 (2022), p. 46.

---

> ### Author Response · Authors · 2024-11-18
>
> Question1.
> In definition 3.4, $\sigma$ can be any element in $C(\mathcal{X}\times\Omega)$. Does this mean we can deal with deep neural networks by properly setting $\sigma$? In that case, I think this framework is more flexible than other methods like ridgelet transform [1] (in the framework of ridgelet transform, we have to consider the form $\sigma(\left<w,x\right>-b)$). Can we apply this framework of RKBS to show the universality of deep neural networks?
>
> Answer:
> 4.Relationship with deep neural networks: To be honest, we are not yet certain whether the flexible notation used in our integral RKBS can be applied to the theoretical analysis of deep neural networks. As mentioned in point 1, we do not rule out such a possibility. However, whether we consider the hypothesis space or universality of deep neural networks, the fundamental philosophy should involve examining models in the context of infinite width. Additionally, since discrete functions also need to be taken into account, a more detailed approach would likely be required. Thus, at the current stage, we believe it is not feasible. Instead, there are a few well-known studies that aim to model the hypothesis space of deep neural networks. Among them, we would like to introduce the following paper, which models deep neural networks using Deep Integral RKBS [2].
>
> [1] S. Sonoda and N. Murata, "Neural network with unbounded activation functions is universal approximator", Applied and Computational Harmonic Analysis, 43(2): 233-268.
>
> [2] Francesca Bartolucci et al. “Neural reproducing kernel Banach spaces and representer theorems for deep networks”. In: arXiv preprint arXiv:2403.08750 (2024).

---

> ### Author Response · Authors · 2024-11-23
>
> Dear reviewer Ubdx,
>
> Once again, we sincerely appreciate your valuable feedback. We have added further elaboration regarding Weakness1 in Section 3.1, and provided additional explanation related to Weakness2 in the Related Work section. However, regarding the practical aspects, we are unable to provide a mathematically perfect description at this point, so we intend to substitute the explanation mentioned above. Additionally, we have addressed the minor comments you raised and uploaded the revised version. Thank you once again for giving us this valuable opportunity.

---

> > ### Comment · Reviewer_Ubdx · 2024-11-24
> >
> > Thank you for the response. The connection between the proposed decomposition and the multiple kernel methods is interesting. I have a question about this point. In practical cases, we use a finite number of kernels for multiple kernel methods, but neural networks are also represented by a finite number of weight parameters and the finite sum instead of the integral. Does this mean if we adopt a certain discretization of the integral and obtain a practical neural network, then it is equivalent to a kernel machine with certain multiple kernels?

---

> > > ### Author Response · Authors · 2024-11-24
> > >
> > > Thank you so much for your response!
> > > We think this is truly an excellent question, and we have learned a lot thanks to you. First, what we intuitively showed in Proposition 5.2 is that neural networks have greater expressive power than kernel machines built with multiple kernels. The part you mentioned seems to be asking whether the reverse holds true in practical scenarios. In many cases, machine learning theory assumes an infinite-dimensional vector space as the hypothesis space. However, when we use a fixed, finite number of neurons, the hypothesis space becomes finite-dimensional. As you pointed out, in practical situations, instead of using an integral representation $f(x) = \int_{\Omega} \sigma(x, w) d\mu(w)$, we use a discretized finite sum with $m$ fixed neurons $$\text{Equation 1}: f(x) = \sum_{i=1}^{m} \eta_{i} \sigma(x, w_{i}) $$ to represent neural networks. If we train using $m$ fixed neurons, the function space representable by Equation 1 would be a finite-dimensional vector space. If we use $\\{\sum_{i=1}^{m} \eta_{i} \delta_{w_{i}}: \eta_{i}, w_{i} \in \mathbb{R}\\}$ as our feature space instead of the measure space $\mathcal{M}(\Omega)$ and develop the discussion further, we might show that models built with a finite number of certain multiple kernels are equivalent to neural networks with $m$ fixed neurons. Furthermore, the situation you described is akin to a scenario where we use Extreme Learning Machines [1] (finite-dimensional) instead of Random Fourier Features [2] (infinite-dimensional) for training a model. We agree that exploring this area further is indeed interesting, and we appreciate the opportunity to continue this dialogue.
> > >
> > > [1] Huang, G.-B., Zhu, Q.-Y., & Siew, C.-K. Extreme learning machine: theory and applications. Neurocomputing, 70(1-3), 489-501, (2006).
> > >
> > > [2] Ali Rahimi and Benjamin Recht. “Random features for large-scale kernel machines”. In: Advances in neural information processing systems 20 (2007).

---

> > > > ### Comment · Reviewer_Ubdx · 2024-11-25
> > > >
> > > > Thank you very much for your interesting and insightful comments. I understand that if we consider the case where the neural networks are represented as a finite sum, the corresponding models in RKHSs shoud also be represented by a finite sum of vectors. But if we use the representer theorem, a model in RKHS is often represented by a finite sum of feature maps even if we do not use any approximation techniques. Is this fact related to the understanding of the connection between the neural networks with a finite number of neurons and multiple kernel machines?

---

> ### Author Response · Authors · 2024-11-26
>
> First of all, we sincerely thank you again for your kind responses and questions.
> We speculate that when considering a model fixed with a finite number of $m$ neurons, the Representer Theorem itself may not provide much useful information. As you know, the intuitive implication of the Representer Theorem is that we only need to consider a finite number of neurons (or a finite sum of kernel functions) instead of an infinite number. Nevertheless, if the number of neurons $m$ is overwhelmingly larger than the number of data points $n$ $(m>>n)$, it seems possible to make meaningful observations.
>
> If the question is not about a model with a fixed
> $m$ neurons but instead about applying the Representer Theorem to the entire integral RKBS
> $\mathcal{F}\_{\sigma}(\mathcal{X},\Omega)$ and its relationship to our decomposition, then this is closely related to the problem we mentioned as a topic of interest in our future work. We believe that since the Representer Theorem determines the properties of the hypothesis space for solving a machine learning problem with a given finite dataset, exploring whether these properties allow us to select a meaningful finite index set from our decomposition $\mathcal{F}\_{\sigma}(\mathcal{X},\Omega)=\sum\_{i\in I}\mathcal{L}\_{\sigma}(\mu\_{i})$ would be an interesting question.
>
> However, we hold a slightly negative view regarding this approach. This is because, even if the space is constrained in this way, it would likely become highly data-dependent. The bottom-up approach that we have consistently mentioned can be seen as an alternative concept to the method described above. In this regard, there is some remarkable research. Specifically, if one defines an integral RKBS based on the ridgelet transform form $g(<x,\hat{w}>-b)$
> and uses a sufficiently good activation function, it has been shown that there exists an RKHS containing the integral RKBS ([1]). We believe this provides additional potential for understanding integral RKBS from a top-down perspective.
>
> It seems this is related to the question you mentioned in Weakness 3. We apologize for not addressing this point when responding to Weakness 3. Discussions with you have been incredibly helpful and delightful for us. Please feel free to ask further questions if you have any, and we will do our best to answer them to the best of our knowledge.
>
>
> [1] Schölpple, M. and Steinwart, I. (2023). Which Spaces can be Embedded in Reproducing Kernel Hilbert Spaces?. arXiv preprint arXiv:2312.14711.

---

> > ### Comment · Reviewer_Ubdx · 2024-11-26
> >
> > Thank you very much for your response. It helped me to understand the proposed framework well. Based on the rebuttal and the additional discussions, I updated my score.

---

> > > ### Author Response · Authors · 2024-11-26
> > >
> > > Thank you so much for taking the time to review our submission. We truly appreciate it!

---

### Official Review · Reviewer_UkFb · 2024-11-02

**Soundness:** 2
**Presentation:** 2
**Contribution:** 1
**Rating:** 5
**Confidence:** 2

**Summary:**

This work studies the reproducing kernel Banach spaces (RKBSs). Specifically, it shows that the integral RKBSs can be decomposed into a sum of a set of RKBSs each defined based on a different measure. It then presents an application of the decomposition result that the RKHSs are contained in the RKBSs.

**Strengths:**

- This work gives a thorough presentation of the related concepts to the sum of the RKBSs proposed here.
- This work shows a nice property that the sum of the feature spaces is compatible with the sum of the RKBSs.

**Weaknesses:**

- The related work section is not informative. In particular, Section 1.1 does not introduce what are the advantages and, importantly to this paper, the limitations of RKHS, and it does not address the previous literature on RKBS nor what questions the literature has solved with RKBS. Also, it does not provide any motivation for the results presented in this work. It is mainly just a list of abbreviated references.
- This work asks the questions to address at the end of page 1 that it aims to decompose the integral RKBS into more fundamental blocks. But it does not touch on the motivation behind and what results one can get with this decomposition.
- Around 5 of the 8 pages are about the definitions or restating results in previous literature. It would be great if this work could spend some space on (1) the potential benefits of their results, (2) takeaway messages about RKBS, (3) technical difficulties encountered and solved, and (4) novel mathematical tools and techniques that are of independent interest. It is otherwise unclear what would be the central contribution of this work.

**Questions:**

- Please see weaknesses above.
- What is $\mathcal{S}$ in the diagram in Figure 1?
- What are some potential applications of the results presented in this work? What are some specific examples of machine learning tasks or theoretical problems where the RKBS decomposition might provide advantages over RKHS approaches?
- What does $2$ mean in $\sum_{i\in [n]}^2$ in Section 5?

---

> ### Author Response · Authors · 2024-11-19
>
> Thank you for your valuable feedback. We acknowledge that our submission did not adequately explain the contribution and motivation behind our work. We will make sure to clarify these points to the best of our ability.
>
> Weakness1.
> The related work section is not informative. In particular, Section 1.1 does not introduce what are the advantages and, importantly to this paper, the limitations of RKHS, and it does not address the previous literature on RKBS nor what questions the literature has solved with RKBS. Also, it does not provide any motivation for the results presented in this work. It is mainly just a list of abbreviated references.
>
> Answer:
> We definitely agree that our explanation in the related work section was insufficient. In previous studies ([1], [2]), various methods were proposed to define and explain the hypothesis space of neural networks. However, these studies were unable to show the existence of solutions in machine learning using the Representer theorem for neural networks. In [3], the authors revealed that the spaces known as the hypothesis space of one-layer neural networks are special Banach spaces that always contain the pointwise convergence topology, which we consider to be the minimal condition for defining a hypothesis space (function space) in machine learning. They defined this space as integral RKBS and used it to prove the Representer theorem for one-layer neural networks. However, proving the existence of a solution does not necessarily lead to finding a specific algorithm. To address this, in this study, we introduced a methodology for decomposing integral RKBS into a family of more manageable (separable) RKBSs, such as $\mathcal{L}\_{\sigma}^{1}(\mu\_{i})$, while preserving the RKBS structure. This approach is done in a way that preserves the structure of the hypothesis space (RKBS structure), which was not visible in previous studies ([4],[5]). We will also add the relevant content you pointed out in Section 1.1. Thank you!
>
> Weakness2.
> This work asks the questions to address at the end of page 1 that it aims to decompose the integral RKBS into more fundamental blocks. But it does not touch on the motivation behind and what results one can get with this decomposition.
>
> Answer:
> Intuitively, it can be explained using the diagram below.
> $$
> \\{\text{ family of singular probability measures } \mu\_{i} \text{ on } \Omega\\} \rightarrow \sum\_{i\in [n]}^{p}\mathcal{L}\_{\sigma}^{p}(\mu\_{i}) \overset{\text{distance}}{\hookrightarrow} \mathcal{F}\_{\sigma}(\mathcal{X},\Omega) = \sum\_{i\in I}^{1}\mathcal{L}\_{\sigma}^{1}(\mu\_{i})
> $$
> As mentioned in Weakness 1, although the Representer theorem for integral RKBS has been proven, no specific algorithm that guarantees the existence of a solution has been developed (and this is not obtainable with the algorithms we actually use for learning). Therefore, we first needed to decompose the integral RKBS while maintaining the structure of RKBS, and we have shown this. As you know, based on the theories from [1] and [6], we can model a one-layer neural network with infinite width by fixing the input-layer parameters chosen from a probability measure $\pi$ and updating only the layer-output parameters, which corresponds to  $\mathcal{L}\_{\sigma}^{2}(\pi)$ in our paper. Therefore, a concrete multiple kernel sum algorithm with the hypothesis space $\sum\_{i\in [n]}^{2}\mathcal{L}\_{\sigma}^{2}(\mu_{i})$ exists. Ultimately, our goal is to develop an algorithm that guarantees the existence of a solution for one-layer neural networks in a bottom-up manner. This could be achieved by finding and minimizing a new distance, as illustrated in the diagram above.
>
> [1] Francis Bach. “Breaking the curse of dimensionality with convex neural networks”. In: The Journal of Machine Learning Research 18.1 (2017), pp. 629–681.
>
> [2] Chao Ma, Lei Wu, et al. “The Barron space and the flow-induced function spaces for neural network models”. In: Constructive Approximation 55.1 (2022), pp. 369–406.
>
> [3] Francesca Bartolucci et al. “Understanding neural networks with reproducing kernel Banach spaces”. In: Applied and Computational Harmonic Analysis 62 (2023), pp. 194–236.
>
> [4] Len Spek, Tjeerd Jan Heeringa, and Christoph Brune. “Duality for neural networks through reproducing kernel Banach spaces”. In: arXiv preprint arXiv:2211.05020 (2022).
>
> [5] E Weinan and Stephan Wojtowytsch. “Representation formulas and pointwise properties for Barron functions”. In: Calculus of Variations and Partial Differential Equations 61.2 (2022), p. 46.
>
> [6]Ali Rahimi and Benjamin Recht. “Random features for large-scale kernel machines”. In: Advances in neural information processing systems 20 (2007).

---

> ### Author Response · Authors · 2024-11-19
>
> Weakness3.
> Around 5 of the 8 pages are about the definitions or restating results in previous literature. It would be great if this work could spend some space on (1) the potential benefits of their results, (2) takeaway messages about RKBS, (3) technical difficulties encountered and solved, and (4) novel mathematical tools and techniques that are of independent interest. It is otherwise unclear what would be the central contribution of this work.
>
> Answer:
> We would like to intuitively explain the contributions of our research. I would like to emphasize that all the content in our propositions and theorems that does not reference existing literature is new theory that we have developed. The specific details are as follows.
>
> Regarding Proposition 3.5:
> We have shown that the bounded operator $A:\mathcal{M}(\Omega)\rightarrow C(\mathcal{X})$ is a compact operator. From this fact, we deduce that the hypothesis space of one-layer neural networks, the integral RKBS, is strictly smaller than the space of continuous functions $C(\mathcal{X})$. To the best of our knowledge, this result has not been previously established in the literature. This implies that while we know from the Universal Approximation Theorem that one-layer neural networks can approximate any continuous function, the hypothesis space of one-layer neural networks cannot cover the entire space of continuous functions. In other words, while the target function we aim to find through the neural network can approximate all continuous functions, an arbitrary continuous function cannot be the target function we are working with.
>
> Regarding Proposition 3.7: The sum of RKHSs (as established in classical results by Aronszajn [1]) has generally only been justified for finite sums. To address this, we used the characterization theorem from [2]. The infinite sum we propose can only be justified by the feature map $\mathbf{s}$ and RKBS map $\mathcal{S}$ defined using equation (2.1) line 132-133, and to the best of our knowledge, there has been no explicit definition of this in previous studies. This may raise the question of why we need to introduce an infinite sum. Since the hypothesis space we aim to work with in machine learning is an infinite vector space, we cannot handle all elements using concepts like the Hamel basis, and thus, the introduction of concepts like the Schauder basis becomes necessary. In a similar context, we required the infinite sum of RKBSs.
>
> Regarding Proposition 4.2: This theorem shows that for an index set $I$ with arbitrary cardinality, the direct sum structure of the feature space is compatible with the sum structure of the RKBS defined in Proposition 3.7. While the result is easily understandable for finite sums, there had been no known results for infinite sums. Since this is stated quite generally, we believe it can be extended to treat other Banach spaces, such as Sobolev spaces, or metrizable locally convex spaces like Fréchet spaces, as feature spaces. Additionally, we believe this theorem strengthens the philosophy that understanding the feature space allows for a better understanding of the hypothesis space.
>
> Regarding Theorem 4.4:
> We have successfully decomposed the integral RKBS, which serves as the hypothesis space for neural networks, while maintaining the RKBS structure, using Proposition 3.7 and Proposition 4.2. As we mentioned in Remark 4.5, this shows that by using spaces such as $\mathcal{L}\_{\sigma}^{1}(\mu\_{i})$ (which are more tractable due to their separability), we can better understand the hypothesis space of neural networks. Furthermore, because we preserved the RKBS structure in the decomposition, we expect that if we can find a kernel learning algorithm for $\mathcal{L}\_{\sigma}^{1}(\mu)$, we could derive a multiple kernel learning algorithm to approximate the approximation power of neural networks.
>
>
> [1] Nachman Aronszajn. “Theory of reproducing kernels”. In: Transactions of the American mathematical society 68.3 (1950), pp. 337–404.
>
> [2] Patrick L Combettes, Saverio Salzo, and Silvia Villa. “Regularized learning schemes in feature Banach spaces”. In: Analysis and Applications 16.01 (2018), pp. 1–54.

---

> ### Author Response · Authors · 2024-11-19
>
> Weakness3.
> Around 5 of the 8 pages are about the definitions or restating results in previous literature. It would be great if this work could spend some space on (1) the potential benefits of their results, (2) takeaway messages about RKBS, (3) technical difficulties encountered and solved, and (4) novel mathematical tools and techniques that are of independent interest. It is otherwise unclear what would be the central contribution of this work.
>
> Answer:
>
> Regarding Propositions 5.1 and 5.2:
> We mentioned them as examples where our theorem can be applied. Specifically, we model a well-known RKHS, $\mathcal{L}\_{\sigma}^{2}(\pi)$, which corresponds to infinite-width one-layer neural networks. In this setup, the input-layer parameters are drawn from a distribution and fixed, while only the layer-output parameters are updated during training. Then, we construct a finite sum of such RKHSs, $\sum\_{i\in [n]}^{2}\mathcal{L}\_{\sigma}^{2}(\mu\_{i})$, as a multiple kernel learning framework while preserving their RKHS structure.We show that this sum of RKHSs belongs to the integral RKBS. Intuitively, this means that the elements of the resulting sum of RKHSs model can be represented by a one-layer neural network. While our theorem primarily discusses RKHSs, as noted in Remark 5.3, the results can also be generalized to arbitrary $p$. The key point we want to emphasize is that we propose a methodology for embedding complex spaces composed of sums of RKBSs, which are often not well understood into well-known RKBSs. This approach could be particularly useful for ensemble methods involving multiple complex models.
>
> Regarding previous literature: You did not provide a reference, so we are unsure which work you are referring to. However, we assume that if you are referring to [1], we acknowledge that many of our notations and definitions were inspired by that reference. Nevertheless, we emphasize that we have clearly cited the sources of all the notations we have used. Additionally, we cautiously point out that the union of vector spaces does not generally retain a vector space structure. The methodology we propose for decomposing while preserving the RKBS structure is entirely original to our work.
>
> (1) The potential benefits of their results:
> As shown in the diagram in Weakness 2, we expect to develop an algorithm that approximates the solution guaranteeing existence in the Representer theorem for one-layer neural networks in a bottom-up manner.
>
> (2) Takeaway messages about RKBS:
> When we define the hypothesis space (function space) in machine learning, we consider completeness and pointwise convergence as minimal assumptions for the properties of the function space. This is because, when approximating a target function through the metric (or topology) of the hypothesis space in machine learning, the approximating function and the target function must at least pointwise converge during the learning process. The key to using RKBS is that, through the characterization theorem (Theorem 3.3), the hypothesis space (RKBS) can always be thought of as a triple consisting of feature space, feature map, and RKBS map (i.e., ($\Psi$, $\psi$, $A$) in our submission). This allows us to understand the hypothesis space once we understand the feature space.
>
> (3) Technical difficulties encountered and solved:
> To handle infinite sums, we carefully verified and introduced related concepts ourselves in the Preliminaries section. More specifically, we did our best to introduce summability and several isometric isomorphisms, which allowed us to apply related concepts freely throughout our proof process.
>
> (4) Novel mathematical tools and techniques that are of independent interest:
> One of the isometric isomorphisms we used, namely equation (2.4), can be thought of as a simpler version of Kakutani's theorem, which classifies Abstract L-spaces. We believe that this theorem itself is quite interesting. Moreover, as mentioned earlier, we appropriately used summability and the isometric isomorphisms we introduced, as needed in our proofs. For example, in lines 650-664 of Proposition 4.2, we successfully transformed a norm defined by the double infimum using the derived properties of summability, even though we did not directly mention it. These methods are not necessarily obvious, and we think they are quite novel.
>
> [1] Len Spek, Tjeerd Jan Heeringa, and Christoph Brune. “Duality for neural networks through reproducing kernel Banach spaces”. In: arXiv preprint arXiv:2211.05020 (2022).

---

> ### Author Response · Authors · 2024-11-19
>
> Question1.
> What is $\mathcal{S}$ in the diagram in Figure 1?
>
> Answer:
> The sum notation $\sum\_{i\in I}^{p}\mathcal{B}\_{i}$ of the RKBS in Proposition 4.2 is not defined in a simple manner. It is, in fact, defined only through Proposition 3.7 and lines 324-329. Specifically, we could only define it using the RKBS triple $(\bigoplus\_{i\in I}^{p}\mathcal{B}\_{i},\mathbf{s},\mathcal{S})$ for the sum. As reviewer pointed out, we recognize that Proposition 3.7 was not stated as clearly as it should have been in our submission, which may have led to some confusion. We will make sure to rewrite and upload the content of Proposition 3.7 in a clearer manner to address this issue.
>
> Question2.
> What are some potential applications of the results presented in this work? What are some specific examples of machine learning tasks or theoretical problems where the RKBS decomposition might provide advantages over RKHS approaches?
>
> Answer:
> Since we are not aware of any methods to express the hypothesis space of neural networks using RKHS, we develop the theory in the context of RKBS, which does not have an inner product structure. Regarding potential applications, we believe we can attempt the direction mentioned in Weakness 2. More specifically, while we know concrete algorithms to find solutions guaranteed by the Representer theorem for RKHSs such as $\mathcal{L}\_{\sigma}^{2}(\pi)$, there are no concrete algorithms to find solutions for other
> $p$-norm RKBS $\mathcal{L}\_{\sigma}^{p}(\pi)$, even though their existence is guaranteed by the Representer theorem. Since the $p$-norm RKBS $\mathcal{L}\_{\sigma}^{p}(\pi)$ has the desirable property of being separable, we can explore specific algorithms in this direction. Ultimately, we aim to find algorithms for the integral RKBS in a bottom-up manner using this approach.
>
> Question3.
> What does $2$ mean in $\sum_{i\in [n]}^{2}$ in Section 5?
>
> Answer:
> There can be various concepts of summation for spaces, but in our paper, we consistently describe the summation notation we wish to use. Therefore, the notation $\sum\_{i\in [n]}^{2}$ used in Section 5 refers to RKHS, which corresponds to the sum of the RKBS defined in lines 324-329. The reason we use $p=2$ is that the sum of the RKBS derived from the 2-norm direct sum $\bigoplus\_{i\in [n]}^{2}$ results in an RKHS. In other words, $\sum\_{i\in [n]}^{2}\mathcal{L}\_{\sigma}^{2}(\mu\_{i})$ is always a Hilbert space, whereas $\sum\_{i\in [n]}^{1}\mathcal{L}\_{\sigma}^{2}(\mu\_{i})$ is generally not a Hilbert space.
>
>
> Thank you for taking the time to read our work. Your review has been immensely helpful to us.

---

> ### Author Response · Authors · 2024-11-23
>
> Dear Reviewer UkFb,
>
> Your valuable feedback has been incredibly helpful to us. In response to Weakness1 you mentioned, we have added further elaboration on the direction of our research in relation to existing studies. Regarding Question1 and Question3, we have provided additional explanations of the relevant definitions in the main text of our revised version.. Once again, we sincerely appreciate your thoughtful review.

---

> > ### Comment · Reviewer_UkFb · 2024-11-25
> >
> > Thank you for your clarifications. I see indeed parts of the draft have been rewritten and better explained. I have raised my score accordingly.

---

> > > ### Author Response · Authors · 2024-11-25
> > >
> > > We sincerely appreciate your thoughtful review and valuable discussion.

---

### Official Review · Reviewer_t3f6 · 2024-11-03

**Soundness:** 3
**Presentation:** 3
**Contribution:** 3
**Rating:** 6
**Confidence:** 3

**Summary:**

The authors define the sum of RKBSs using a characterization theorem, investigate its compatibility with the direct sum of feature spaces, and decompose the integral RKBS $ F_\sigma(X, \Omega) $ into the sum of $p$-norm RKBSs $\{L^1_\sigma(\mu_i)\}_{i \in I}$. This study enhances the structural understanding of the integral RKBS class, offering theoretical insights that can help analyze the performance of neural networks by decomposing complex function spaces into simpler, manageable components.

**Strengths:**

The paper introduces an innovative framework for decomposing integral RKBSs, offering a novel interpretation of one-layer neural networks. This approach is unique in its use of Banach spaces and their decomposition to analyze function spaces, advancing the existing understanding of RKBSs.

The decomposition of RKBSs has significant implications for the analysis of neural networks, especially in designing kernel-based learning algorithms. The compatibility between the sum of RKBSs and the direct sum of feature spaces represents a meaningful advancement in understanding how integral RKBSs can be decomposed, which could potentially impact practical applications in machine learning, such as multiple kernel learning.

**Weaknesses:**

The paper could benefit from more illustrative examples to make the abstract mathematical concepts more accessible to a broader audience, particularly those in the machine learning community without a strong background in functional analysis.

The experimental results are limited, and the practical implications of the theoretical findings are not fully demonstrated through empirical evaluation. Including numerical examples or simulations to show the decomposition's effects on real-world neural network performance would significantly improve the paper's practical relevance.

The presentation of some key definitions and theorems is rather dense, making it difficult for readers to follow the logical flow. Providing intuitive explanations alongside formal proofs would help bridge the gap for less mathematically inclined readers.

**Questions:**

1. Could the authors provide a concrete example of how the decomposition of an RKBS improves the understanding or efficiency of neural network analysis? An illustrative example or a simple simulation would greatly help clarify the practical benefits.

2. Would the authors consider adding a numerical evaluation to demonstrate the theoretical claims empirically? This would help bridge the gap between the abstract mathematical results and their practical implications in machine learning.

---

> ### Author Response · Authors · 2024-11-21
>
> Thank you so much for your thoughtful and valuable feedback. We believe your question about whether the mathematical theories in our submission can be effectively evaluated through numerical experiments is particularly insightful and greatly appreciated.
>
> Weakness1.
> The paper could benefit from more illustrative examples to make the abstract mathematical concepts more accessible to a broader audience, particularly those in the machine learning community without a strong background in functional analysis.
>
> Weakness3.
> The presentation of some key definitions and theorems is rather dense, making it difficult for readers to follow the logical flow. Providing intuitive explanations alongside formal proofs would help bridge the gap for less mathematically inclined readers.
>
> Answer: We believe that the two questions are essentially the same, so we will address them simultaneously. As you pointed out, our submission is written in formal mathematical language, which we understand might pose challenges for many readers. While it would have been helpful to include illustrative examples, we aimed to avoid potential misunderstandings that could arise from using non-mathematical language. Additionally, since the subject we are addressing is inherently a mathematical object, this approach seemed unavoidable. We will add further explanations to aid readers' understanding and will upload the revised version.
>
>
> Weakness2.
> The experimental results are limited, and the practical implications of the theoretical findings are not fully demonstrated through empirical evaluation. Including numerical examples or simulations to show the decomposition's effects on real-world neural network performance would significantly improve the paper's practical relevance.
>
> Answer: To begin with, let us explain the motivation behind our study. For the hypothesis space of one-layer neural networks (the integral RKBS class), [1] proved a Representer Theorem in Theorem 3.9. This theorem guarantees the existence of a solution by showing that our target function can be expressed as a finite sum of functions when solving optimization problems in one-layer neural networks using empirical risk minimization (ERM). However, the existence of such a solution does not guarantee the existence of a specific algorithm to find it. Therefore, it is likely that the optimization methods we actually use for neural networks do not find the solution guaranteed by the Representer Theorem. (In fact, we believe that additional dummy parameters considered in practice lead to an increase in the generalization error compared to the solution guaranteed by the Representer Theorem.) In the case of kernel methods using RKHS (e.g., kernel ridge regression), the theoretical existence of a solution and the algorithm to find it are clear. We aimed to find a similar methodological approach for neural networks, where both the theoretical existence of solutions and the practical optimization algorithms can be identified, which motivated us to explore this decomposition. However, there are still limitations in conducting numerical analyses at this stage. We will elaborate on this further in the responses to Question 1 and Question 2 below.
>
> [1] Francesca Bartolucci, Ernesto De Vito, Lorenzo Rosasco, and Stefano Vigogna. Understanding neural networks with reproducing kernel Banach spaces. Applied and Computational Harmonic Analysis, 62:194–236, 2023

---

> ### Author Response · Authors · 2024-11-21
>
> Question1.
> Could the authors provide a concrete example of how the decomposition of an RKBS improves the understanding or efficiency of neural network analysis? An illustrative example or a simple simulation would greatly help clarify the practical benefits.
>
> Answer: This seems to be a question related to Section 5 of our submission. We would like to provide a more intuitive explanation for Proposition 5.1. As you may know, $\mathcal{L}\_{\sigma}^{2}(\pi)$ models a one-layer neural network with infinite width, where the input-layer parameters are fixed from the distribution $\pi$ and only the layer-output parameters are learned. This model is an RKHS, and it is likely to represent the space of many kernel method algorithms that we can practically handle ([1],[2]). We have shown that the hypothesis space $\sum\_{i\in [n]}^{2}\mathcal{L}\_{\sigma}^{2}(\mu\_{i})$, which we aim to solve using a multiple kernel algorithm for combinations of all finite singular probability measures, is embedded in the integral RKBS which represent the one-layer neural network hypothesis space. In other words, most of the RKHSs used in kernel methods that we consider are less effective in terms of approximation ability compared to one-layer neural networks.
>
> Question2.
> Would the authors consider adding a numerical evaluation to demonstrate the theoretical claims empirically? This would help bridge the gap between the abstract mathematical results and their practical implications in machine learning.
>
> Answer:
> In Question 1, we explained the intuition behind our theory. However, we currently do not have a way to implement and prove this experimentally. As mentioned in Weakness 2, this is because we do not have an algorithm to optimize the integral RKBS $\mathcal{F}\_{\sigma}(\mathcal{X},\Omega)$ in a way that guarantees the existence of a solution. Since we do know an algorithm (which find a solution guaranteed by the Representer Theorem) to compute $\sum\_{i\in [n]}^{2}\mathcal{L}\_{\sigma}^{2}(\mu\_{i})$, how about comparing the results using the existing method for optimizing one-layer neural networks under square loss? However, this approach would also struggle to prove space size comparisons in practice, and even if the experimental results for one-layer neural networks are good, they would not be reliable due to the reasons mentioned earlier. Instead, our approach suggests future research possibilities for practical directions. The candidate approach is to progressively add $\mathcal{L}\_{\sigma}^{p}(\mu\_{i})$ terms in a bottom-up manner, getting as close as possible to the integral RKBS. In summary, we have conducted research to reduce the gap between theory and practical experiments, and our goal is to find concrete (numerically feasible) algorithms that are guaranteed by the theory. This is the direction we aim to pursue.
>
> [1] Ali Rahimi and Benjamin Recht. “Random features for large-scale kernel machines”. In: Advances in neural information processing systems 20 (2007).
>
> [2] Francis Bach. “Breaking the curse of dimensionality with convex neural networks”. In: The Journal of Machine Learning Research 18.1 (2017), pp. 629–681.

---

> > ### Comment · Reviewer_t3f6 · 2024-11-22
> >
> > Thank you for your explanation. I agree with your argument.

---

> > > ### Author Response · Authors · 2024-11-23
> > >
> > > Dear Reviewer t3f6,
> > >
> > > Once again, we sincerely appreciate your valuable feedback :)

---

### Meta-Review · Area_Chair_zfA3 · 2024-12-26

**Metareview:**

The submission studies reproducing kernel Banach spaces. RKBS arise in connection with infinitely wide neural networks; the paper also posits them as a relevant hypothesis space for learning problems due to their properties of completeness and pointwise convergence. The paper has two main results. The first (Prop 4.2) shows the compatibility of RKBS with the direct sum operation, namely, the sum of RKBS is isometrically isomorphic to an RKBS. The second (Thm 4.4) states that an RKBS in integral form can be written as a sum of L1 spaces, with different feature distributions.

As described below, reviewers initially produced a mixed evaluation of the paper. On the positive side, the paper is mathematically solid, and contributes several results on the structure of reproducing kernel Banach spaces. As the paper notes, these spaces are less studied than reproducing kernel hilbert spaces. Whereas the summability of RKHS captures the hypothesis space associated with fixed first layer features, structural results on RKBS could shed light on networks with varying first layer features. The paper shows that an RKBS in integral form can be decomposed into simpler spaces (L1 spaces).

While the discussion provided useful context for appreciating the paper’s results, the paper would still benefit from a clearer and more accessible discussion of their implications for neural networks, as well as their significance in the broader program of establishing algorithms for solving learning problems in RKBS.

**Additional Comments On Reviewer Discussion:**

Reviewers found the paper mathematically solid, contributing several results on the structure of reproducing kernel Banach spaces. At the same time, several reviewers raised questions regarding the direct implications of this analysis for neural network learning.In particular, reviewers raised the following issues:

- Mathematically dense presentation [t3f6, UkFb]
- Reviewers found the connection neural networks are unclear [t3f6, UkFb, Ubdx, mrzL], and were similarly unclear on the practical implications of the results.

The discussion also clarified the technical novelties of the paper (e.g., showing compactness of the operator M(\Omega)). The discussion also clarified certain higher level aspects of the paper’s contributions. Roughly, there is a separation between RKHS (which correspond to fixed feature methods) and RKBS. Previous work has demonstrated the existence of an optimal solution in RKBS, but we currently lack a corresponding concrete algorithm for computing this solution. Structural results on RKBS could facilitate the development of such an algorithm.

---

### Decision · Program_Chairs · 2025-01-22

Reject